# Pan-genome inversion index reveals evolutionary insights into the subpopulation structure of Asian rice

Yong Zhou[1,2,13], Zhichao Yu[3,13], Dmytro Chebotarov[4,13], Kapeel Chougule[5,13], Zhenyuan Lu[5], Luis F. Rivera[1], Nagarajan Kathiresan[6], Noor Al-Bader[1], Nahed Mohammed[1], Aseel Alsantely[1], Saule Mussurova[1], João Santos[1], Manjula Thimma[1], Maxim Troukhan[7], Alice Fornasiero[1], Carl D. Green[8], Dario Copetti[2], David Kudrna[2], Victor Llaca[9], Mathias Lorieux[10], Andrea Zuccolo[1,11,14] ✉, Doreen Ware[5,12,14] ✉, Kenneth McNally[4,14] ✉, Jianwei Zhang[2,3,14] ✉ & Rod A. Wing[1,2,4,14] ✉

Understanding and exploiting genetic diversity is a key factor for the productive and stable production of rice. Here, we utilize 73 high-quality genomes that encompass the subpopulation structure of Asian rice (*Oryza sativa*), plus the genomes of two wild relatives (*O. rufipogon* and *O. punctata*), to build a pan-genome inversion index of 1769 non-redundant inversions that span an average of ~29% of the *O. sativa* cv. Nipponbare reference genome sequence. Using this index, we estimate an inversion rate of ~700 inversions per million years in Asian rice, which is 16 to 50 times higher than previously estimated for plants. Detailed analyses of these inversions show evidence of their effects on gene expression, recombination rate, and linkage disequilibrium. Our study uncovers the prevalence and scale of large inversions (≥100 bp) across the pan-genome of Asian rice and hints at their largely unexplored role in functional biology and crop performance.

Asian rice (*Oryza sativa*) is a staple cereal crop that has played an essential role in feeding much of the world for millennia[1,2]. Since the world population is expected to increase to approximately 10 billion by 2060–2070[3], the rice community is searching for novel ways to breed new varieties that are nutritious, sustainable, and climate resilient[2]. One source of the raw material required to meet this urgent demand is the standing natural variation that exists in the genomes of the more than 500,000 accessions of rice and its wild relatives deposited in germplasm banks around the world[2], i.e., single nucleotide polymorphisms (SNPs),

[1]Center for Desert Agriculture (CDA), Biological and Environmental Sciences & Engineering Division (BESE), King Abdullah University of Science and Technology (KAUST), Thuwal 23955-6900, Saudi Arabia. [2]Arizona Genomics Institute (AGI), School of Plant Sciences, University of Arizona, Tucson, AZ 85721, USA. [3]National Key Laboratory of Crop Genetic Improvement, Hubei Hongshan Laboratory, Huazhong Agricultural University, Wuhan 430070, China. [4]International Rice Research Institute (IRRI), Los Baños, 4031 Laguna, Philippines. [5]Cold Spring Harbor Laboratory, Cold Spring Harbor, NY 11724, USA. [6]Supercomputing Core Lab, King Abdullah University of Science and Technology (KAUST), Thuwal 23955-6900, Saudi Arabia. [7]Persephone Software, LLC, Agoura Hills, CA 91301, USA. [8]Information Technology Department, King Abdullah University of Science and Technology (KAUST), Thuwal 23955-6900, Saudi Arabia. [9]Research and Development, Corteva Agriscience, Johnston, IA 50131, USA. [10]DIADE, University of Montpellier, CIRAD, IRD, Montpellier, France. [11]Crop Science Research Center (CSRC), Scuola Superiore Sant'Anna, Pisa 56127, Italy. [12]USDA ARS NEA Plant, Soil & Nutrition Laboratory Research Unit, Ithaca, NY 14853, USA. [13]These authors contributed equally: Yong Zhou, Zhichao Yu, Dmytro Chebotarov, Kapeel Chougule. [14]These authors jointly supervised this work: Andrea Zuccolo, Doreen Ware, Kenneth McNally, Jianwei Zhang, Rod A. Wing. ✉e-mail: andrea.zuccolo@kaust.edu.sa; Doreen.ware@usda.gov; k.mcnally@irri.org; jzhang@mail.hzau.edu.cn; rod.wing@kaust.edu.sa

insertions/deletions (INSs/DELs), translocations (TRAs), and inversions (INVs).

Inversions have been reported as an important class of structural variation across all life forms, beginning with the first reported inversion in *Drosophila* in 1921[4], followed by discoveries in fungi[5], bacterial[6], plants[7–12], and animals[13–17]. In many cases, inversions have been found to play important roles in genetic recombination, genome evolution, speciation, and phenotypic variation in these species. For example, in *Drosophila subobscura*, inversions have been hypothesized as a driver of genomic structural evolution due to suppressed recombination in inverted regions that contain adaptive genes[18]. In birds, the great tit (*Parus major*) has a 64 megabase (Mb) inversion that covers more than 90% of chromosome 1A (5% of a 1.2 Gb genome size), which was shown to be involved in speciation[19]. In humans, many INVs have been found to be associated with human diseases, e.g., hemophilia A[20], neurodegenerative diseases[21], autoimmune diseases[22], mental disorders[23], and gene expression abnormalities[15,24]. In plants, INVs have been reported to play roles in, for example - local adaptation[10,25], genome-environment associations[25], gene regulation[10,26,27], flowering time[26], seed germination[26], and fruit shape[27].

In rice and the genus *Oryza*, inversion studies have been limited to small and mid-size inversions (10 bp–500 Kb) as a consequence of the reliance on short-read data for their detection or larger inversions (100 bp–5.5 Mb) via single or a limited number of pairwise genome scans. For example, Wang et al. performed a genome scan of inversions in the *O. sativa* cv. Nipponbare genome (i.e., IRGSP RefSeq) using short-read resequencing data from 453 high-coverage genomes (>20×) from the 3K Rice Genome Project (3K-RGP), and detected $152 \pm 62$ inversions per genome with a size range of $127.1 \pm 19.4$ Kb[28,29]. A phylogenetic analysis of this dataset, including other SV data, demonstrated that SVs could be used to define the population structure of Asian rice[28]. Fuentes et al. investigated the entire 3K-RGP dataset similarly and identified 1,255,033 inversions, with the vast majority (85%) falling in a size range of 10 bp–100 Kb[11]. Using a genome scanning strategy, a scan of the IRGSP RefSeq, plus reciprocal genome alignments to nine Asian rice and two AA-genome wild relatives (i.e., *O. rufipogon* and *O. longistaminata*) confirmed the presence a previously detected ~5 Mb inversion spanning the centromere of chromosome 6 in four *Xian-indica* (*XI*) varieties, relative to four *Geng-japonica* (*GJ*) varieties, as well as two outgroup species[30]. A broader phylogenetic study interrogating two cultivated and 11 wild *Oryza* genomes using SVs resulted in the identification of 12 inversions (i.e., 60–300 Kb) that the authors inferred potentially led to the rapid diversification of the AA genome species within a 2.5 million years (MY) span[31]. Recently, a genome scan of 33 rice genome assemblies identified 954 inversions ranging in size from 100 bp to ~5 Mb[12]. This study identified larger inversions that included seven inversions greater than 1 Mb.

Although these studies combined contributed to a preliminary understanding of inversions in rice, a comprehensive analysis of inversions that utilize ultra-high-quality reference genome sequences that takes into account the population structure of Asian rice, remains uncharted. To reveal a comprehensive understanding of inversions (≥100 bp) and explore their evolutionary impacts in Asian rice, we interrogated a set of 73 high-quality genome assemblies - as a pan-genome proxy - that spanned the K = 15[32] population structure of cultivated Asian rice, plus two additional high-quality de novo assembled genomes from a representative species of the progenitor of Asian rice (*O. rufipogon*) and the BB genome species - *O. punctata*, as outgroups.

Here, we describe and characterize a pan-genome inversion index (PGII) of 1769 non-redundant inversions, of which 1085 are identified in this study. Phylogenetic analysis of the PGII shows that 66 of the 73 genomes can be subdivided across the expected K = 15 subpopulation structure of Asian rice, while the remaining seven fall into two Xian/ Indica clusters that have yet to be characterized. Clustering the index into 1426 inversion clusters shows that most clusters are *O. sativa*

species-specific (n = 885), followed by *O. rufipogon* species-specific (n = 96), *O. punctata* species-specific or AA genome fixed (n = 322), and basal to *O. sativa/O. rufipogon* divergence, or introgression (n = 123). Characterization of subsets of the index shows evidence for both recombination suppression as well as trace patterns of linkage disequilibrium, both of which suggest that some inversions may be under positive selection. Finally, with this index we estimate a population-level inversion rate for Asian rice of 735–749 inversions per million years, which is 16–50 times higher than previous estimates for plants.

## Results

### The 18-genome data package

To investigate the genome inversion landscape of Asian rice from a population structure perspective, we first combined a set of 16 previously published high-quality genomes[32–34] that represent the K = 15 population structure of *O. sativa*, plus the largest *Xian/indica* (*XI*) admixed subpopulation (*XI*-adm: Minghui 63 (MH63)) to create a "Rice Population Reference Panel" (RPRP - https://yongzhou2019.github.io/ Rice-Population-Reference-Panel/). This panel was annotated with a uniform pipeline to minimize methodological artifacts using RNA-Seq and Iso-Seq data generated or collected from all 16 genomes as evidence (Table 1, Supplementary Tables 1–3, Supplementary Fig. 1, Supplementary Data 1, and Supplementary Note 1).

To anchor this panel within a phylogenetic context, we long-read sequenced, and de novo assembled two additional genomes from both a representative species of the progenitor of Asian rice - i.e., *O. rufipogon* (AA) and the African BB genome outgroup species - *O. punctata* (Table 1, Supplementary Table 4, and Supplementary Fig. 2). Both species are diploid and have similar genome sizes as Asian rice. These genomes were assembled to a similar quality as the RPRP (i.e., BUSCO > 95%, number of gaps <50, ContigN50 > 10 Mb) (Supplementary Table 4 and Supplementary Note 1).

All 18 genomes and their annotations are henceforth referred to as the "18-genome data package" (See "18-genome data package" in the Supplementary Note 1 section for a complete description of this data set).

### Creation of a pan-genome inversion index for Asian rice

To ensure the detection of the majority of inversions present in cultivated Asian rice, we selected an additional 57 publicly available long-read genome assemblies (out of 94) for analyses that met similar quality standards present in the 18-genome data package, for a total of 75 genomes (Supplementary Fig. 3, Supplementary Data 2, and Supplementary Note 2).

To detect inversions, we compared pairwise all 74 reference genome assemblies with the IRGSP RefSeq[34], following a workflow as detailed in Supplementary Note 3 (Supplementary Fig. 4 and Supplementary Table 5), and identified a total of 12,141 inversions (≥100 bp) (Supplementary Data 3), 1769 of which were non-redundant (Supplementary Data 4). Finally, since several inversions had overlapping coordinates (i.e., 80% of their length), they were clustered to obtain a set of 1426 clustered inversions (Supplementary Data 4). We then validated a subset of 264 inversions from four randomly selected genome assemblies (i.e., LM, NABO, CM, and MH63) using PacBio long-reads, and found that 97.7% of the inversion breakpoints could be detected (Supplementary Table 6), thereby confirming the high accuracy of our detection strategy.

To determine if the use of 75 genomes was sufficient to call the majority of 100 bp or greater inversions in the pan-genome of Asian rice, we performed a permutation test (n = 1000), and found that the use of ~60 genomes was sufficient to capture the majority of inversions with allele frequencies greater than 2 out of 75 genomes (Fig. 1a, b). These results demonstrate that employing 75 genomes, that bridge the K = 15 population structure of cultivated rice, can yield a robust pan-genome inversion index for Asian rice.

## Table 1 | Assembly and annotation statistics of the 18-genome data package

| Acronyms | Variety name/Accession ID | Species/genome | Subpopulation[a] | GenBank accession | Assembly size (Mb) | Repeat | Annotated loci[b] |
|---|---|---|---|---|---|---|---|
| IRGSP | IRGSP-1.0/NIPPONBARE | *O. sativa*-AA (Asian rice) | *GJ*-temp | GCA_001433935.1 | 373.25 | 51.79% | 37,140 |
| CMeo | CHAO MEO::IRGC 80273-1 | *O. sativa*-AA (Asian rice) | *GJ*-subtrp | GCA_009831315.1 | 376.86 | 47.41% | 36,601 |
| Azu | Azucena | *O. sativa*-AA (Asian rice) | *GJ*-trop1 | GCA_009830595.1 | 379.63 | 52.56% | 36,623 |
| KeNa | KETAN NANGKA::IRGC 19961-2 | *O. sativa*-AA (Asian rice) | *GJ*-trop2 | GCA_009831275.1 | 380.76 | 50.30% | 36,609 |
| ARC | ARC 10497::IRGC 12485-1 | *O. sativa*-AA (Asian rice) | *cB* | GCA_009831255.1 | 378.46 | 49.50% | 36,423 |
| PR106 | PR 106::IRGC 53418-1 | *O. sativa*-AA (Asian rice) | *XI*–1B2 | GCA_009831045.1 | 391.18 | 50.45% | 36,405 |
| MH63 | Minghui 63 | *O. sativa*-AA (Asian rice) | *XI*-adm | GCA_001623365.2 | 387.43 | 53.41% | 38,047 |
| IR64 | IR 64 | *O. sativa*-AA (Asian rice) | *XI*–1B1 | GCA_009914875.1 | 386.7 | 53.55% | 36,065 |
| ZS97 | Zhenshan 97 | *O. sativa*-AA (Asian rice) | *XI*–1A | GCA_001623345.2 | 387.33 | 53.41% | 37,651 |
| Lima | LIMA::IRGC 81487-1 | *O. sativa*-AA (Asian rice) | *XI*–3A | GCA_009829395.1 | 392.63 | 48.93% | 36,217 |
| KYG | KHAO YAI GUANG::IRGC 65972-1 | *O. sativa*-AA (Asian rice) | *XI*–3B1 | GCA_009831295.1 | 393.74 | 53.93% | 36,212 |
| GoSa | GOBOL SAIL (BALAM)::IRGC 26624-2 | *O. sativa*-AA (Asian rice) | *XI*–2A | GCA_009831025.1 | 391.77 | 51.01% | 36,222 |
| LiXu | LIU XU::IRGC 109232-1 | *O. sativa*-AA (Asian rice) | *XI*–3B2 | GCA_009829375.1 | 392.03 | 51.66% | 36,378 |
| LaMu | LARHA MUGAD::IRGC 52339-1 | *O. sativa*-AA (Asian rice) | *XI*–2B | GCA_009831355.1 | 390.2 | 51.25% | 36,299 |
| N22 | N22 (N 22::IRGC 19379-1) | *O. sativa*-AA (Asian rice) | *cA1* | GCA_001952365.2 | 382.95 | 52.82% | 36,262 |
| NaBo | NATEL BORO::IRGC 34749-1 | *O. sativa*-AA (Asian rice) | *cA2* | GCA_009831335.1 | 383.72 | 50.64% | 36,196 |
| *O. ruf* | *O. rufipogon* PNG91-7::IRGC 106523-1 | *O. rufipogon*-AA (Asian rice progenitor) | – | GCA_023541355.1 | 462.58 | 56.54% | – |
| *O. pun* | *O. punctata*:: IRGC 105690 | *O. punctata*-BB (Outgroup) | – | GCA_000573905.2 | 422.39 | 53.21% | – |

Previously published genomes and annotations are cited.

[a]Subpopulations: *GJ = Geng/Japonica* where trop = tropical, subtrp = subtropical; *cB = circum/Basmati*; *XI = Xian/Indica*; *cA = circum-Aus*.

[b]Gene annotations can be obtained from Gramene (https://www.gramene.org/) or the RPRP website (https://yongzhou2019.github.io/Rice-Population-Reference-Panel/data/).

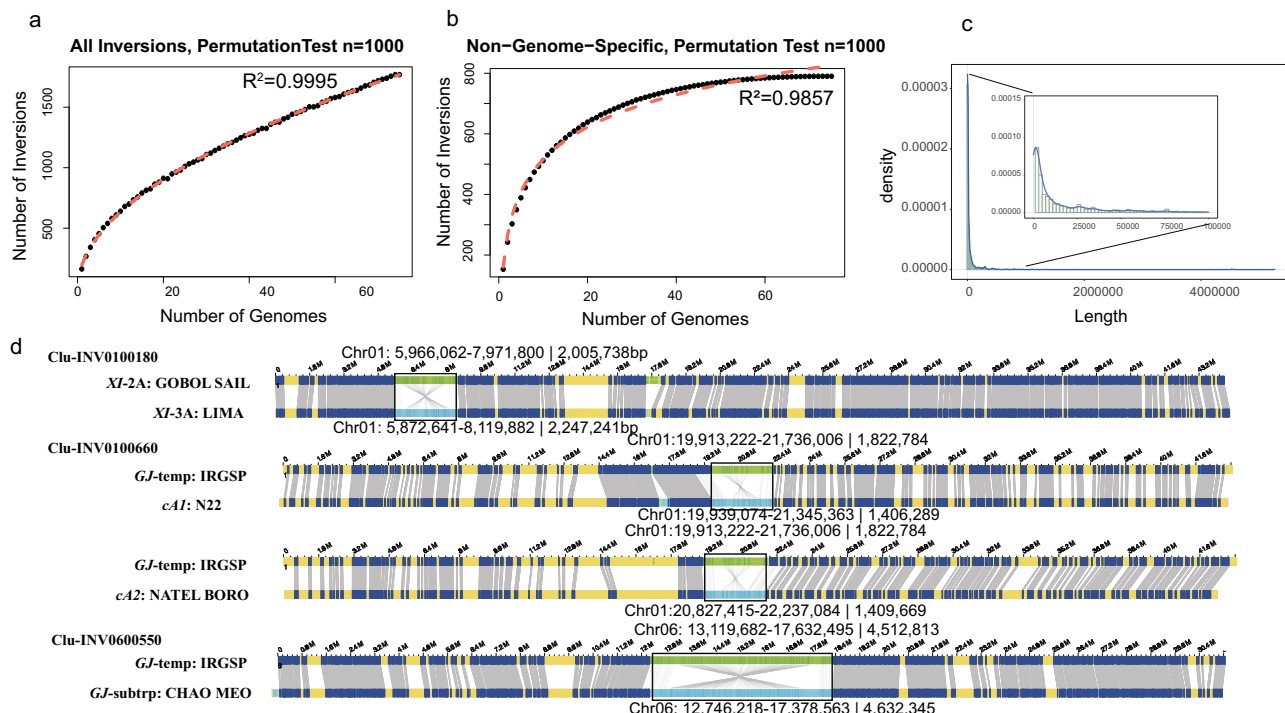

**Fig. 1 | Rice inversion index summary. a**, **b** Resampling permutation test to identify the relationship between the number of genomes and all inversions and shared (non-genome specific) inversions, respectively; **c** Density of inversion lengths; **d** Bionano validation of inversions larger than 1 Mb, i.e., Clu-INV0100180, Clu-INV0100660, and Clu-INV0600550. In each panel, the top line shows the optical map used as a reference, the bottom line shows the genome assembly of the variety with the inversion. Gray lines connect restriction sites that are aligned (blue regions), while yellow segments show unaligned regions. Black boxes highlight the position each inversion. Source data are provided as a Source Data file.

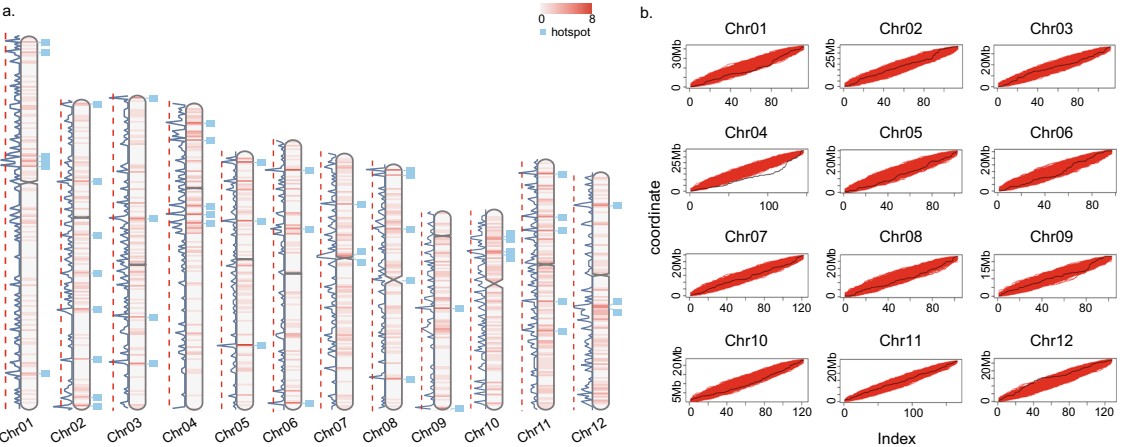

**Fig. 2 | Genome-wide inversion distribution. a** Chromosome distribution of the pan-genome inversion index, and inversion hotspots. Chromosome heatmaps in the middle along the chromosome represent the density of inversions. The line on the left of each chromosome represents the number of inversions per 200 Kb window. The red dotted line cut off is the top 2% number of inversions in per 200 Kb window. The blue boxes on the right of each chromosome represents the inversion hotspot regions. **b** The Kolmogorov–Smirnov (KS) test for inversion uniformity distribution across the 12 rice chromosomes. The black line is actual inversion distribution, and the red line is the 10,000th uniformly distributed simulation. Source data are provided as a Source Data file.

Our inversion index was compared with previous studies in rice, i.e., 3K-RGP[11] and the pan-genome study of 33 rice genomes[12]. Inversions were treated as identical if they matched the following two criteria: 1) the inversion length difference was smaller than 200 bp, and 2) the differences between the coordinates of breakpoints of inversions were smaller than 100 bp. In doing so, we found that of the 1769 nonredundant inversions identified, 38.6% have already been reported (Supplementary Data 4).

As expected, more inversions were observed when we compared the BB genome outgroup (*O. punctata*) than AA genomes to the IRGSP RefSeq: 337 (total length = 15.16 Mb) (Supplementary Data 3), followed by the AA genome outgroup (*O. rufipogon*): 230 (total length = 12.95 Mb). On average, each *O. sativa* genome was found to contain 164 inversions, ranging from 33 (TG22) to 230 (YX1) (Supplementary Data 3). We found a larger number of inversions (i.e.,164–230) when comparing the *O. sativa XI*-subgroup genomes to the *GJ*-temp: IRGSP RefSeq than when comparing the *O. sativa GJ*-subgroup genomes (33–201 inversions) to the same reference, which is consistent with the long divergence time estimate between the *XI* and *GJ* varietal groups (i.e., 360,000–400,000 years[35]).

The total length of inversions from one genome to another ranged from 14.95 Mb (*cA2*: NATEL BORO) to 255 Kb (*GJ*-temp: TG19) (Supplementary Data 3) and totaled 117.75 Mb across all 75 genomes tested. We identified 470 clustered inversions shorter than 1 Kb and 11 clustered inversions greater than 1 Mb relative to the IRGSP RefSeq (Fig. 1c and Supplementary Data 4). These extra-large inversions (>1 Mb) were found on seven chromosomes (i.e., 1, 4, 5, 6, 8, 10, and 11) and included seven of which were previously reported (i.e., Clu-INV0600550 (4.5 Mb)[30], Clu-INV0800800 (1.1 Mb)[36], Clu-INV0100660 (1.8 Mb)[12], Clu-INV0500270 (3.1 Mb)[12], Clu-INV0800450 (5.2 Mb)[12], Clu-INV0800470 (1.2 Mb)[12], Clu-INV110070 (1.1 Mb)[12], and four were identified in this study (i.e., Clu-INV0100180 (2 Mb), Clu-INV0400050 (1.3 Mb), Clu-INV0400650 (4.3 Mb), and Clu-INV1000940 (1.3 Mb)). Three of the 11 clustered inversions (i.e., Clu-INV0100180, Clu-INV0100660, and INV0600550) were validated with available Bionano optical maps (Fig. 1d), while the remaining eight were not due to the lack of Bionano data.

### Chromosomal distribution of the pan-genome inversions index

To determine if the inversions detected were uniformly distributed across all chromosomes, we tested for uniformity using the Kolmogorov–Smirnov (KS) test, and found that only one chromosome, chromosome 4, deviated significantly from uniformity (i.e., $p = 3e{-}04$; Fig. 2a, b, and Supplementary Data 5). To check if inversion size played a role in this chromosome 4 anomaly, we repeated the KS test by classifying inversions according to their lengths (<1 Kb, 1–5 Kb, 5–10 Kb, and >10 Kb). We found no significant deviation from uniformity except for the >10 Kb inversion class on chromosome 4 (Supplementary Fig. 5 and Supplementary Data 5), where inversions are mainly concentrated near the centromere on both the short and long arms of chromosome 4. These results demonstrate that the inversions detected appear evenly distributed genome-wide, with one exception.

To search for inversion hotspots, we performed a 200 Kb window-based analysis across all chromosomes. We defined the top 2% of all windows with the highest frequency of inversion start coordinates as hotspots. This analysis revealed 47 putative hotspots, including 239 independent inversions where 12 inversions overlapped with centromeres (on chromosomes 7 and 8) but none with telomeres (Fig. 2a and Supplementary Data 6).

### Phylogenetic analysis of the pan-genome inversion index

To determine if the pan-genome inversion index could be used to phylogenetically distinguish each of the 75 high-quality genomes into the expected $K = 15$ subpopulation structure, we used the unweighted pair group with arithmetic mean distance tree method, with the *O. punctata* (BB) and *O. rufipogon* (AA) genomes as outgroups (Fig. 3). As shown in Fig. 3, 66 of the 73 genomes could be subdivided across the expected subpopulation structure of Asian rice, while the remaining seven fell into two *XI* clusters that have yet to be characterized.

### Inversion rate estimations for Asian rice

It was estimated that the AA genomes of the *Oryza* diverged from the BB genome type about 2.5 million years ago (MYA)[2], which equates to an inversion rate of 67.4 inversions per million years (Table 2 and Fig. 4a). If we use the inferred AA genome diversification (i.e., *O. sativa GJ*-temp IRGSP vs. *O. rufipogon*) rate of ~0.50 net new species/million years[2,31], then the estimate is 230 inversions per MY (Table 2 and Fig. 4a).

Regarding *O. sativa* subpopulations, it was estimated that temperate *japonica* (*GJ*-temp) first diverged from proto-*japonica* about 14,200 years ago[37], thus we used this population divergence time for comparisons within *Geng/japonica*. For the computation of time to a MRCA for a pair of genomes, we added the expected time to coalesce

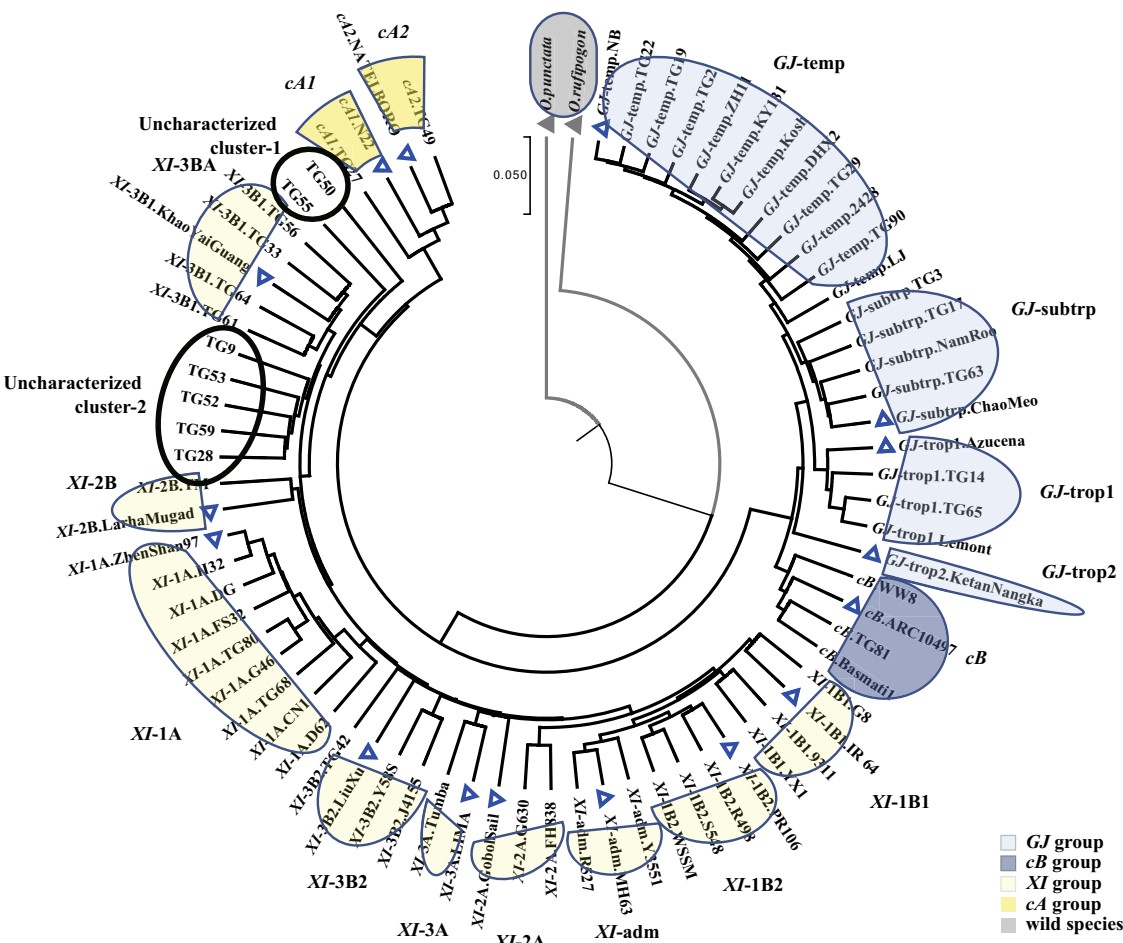

**Fig. 3 | Phylogenetic tree of 75 high-quality rice genomes using the pan-genome inversion index.** Phylogenetic relationships of the 75 high-quality genomes used to create the pan-genome inversion index, inferred using the UPGMA method (unweighted pair group method with arithmetic mean[95]). The tree is drawn to scale, with branch lengths in the same units as those of the evolutionary distances used to infer the phylogenetic tree. Evolutionary analyses were conducted in MEGA7[96]. Two wild relatives (*O. rufipogon* and *O. punctata*) were heighted with light gray arc. Subpopulations from *GJ*, *cB*, *XI*, and *cA* groups were highlighted with light blue, dark blue, light yellow and dark yellow, respectively. Two uncharacterized *XI* sub-populations are shown with a black circle. The distance matrix of SNP and INV polymorphisms was significantly correlated (Mantel test, simulation with $n = 999,999$, $r = 0.79$, $p = 1e-6$). Source data are provided as a Source Data file.

**Table 2 | Inversion rate estimations**

| Comparison | Time to MRCA (MY) | Number of inversion polymorphisms | | Inversion mutation rate (IR) |
|---|---|---|---|---|
| | | Raw | Usable[a] | Number of inversions per million years |
| *O. punctata* - NB | 2.5 | 323 | 323 | 64.6 |
| *O. rufipogon* - NB | 0.5 | 230 | 230 | 230 |
| *GJ*-temp vs other *GJ* (average) | 0.0142 | 82.96 | 21.27 | 749 |
| Average of pairs of *GJ* genomes | 0.0142 | | 20.86 | 735 |

*MRCA* most recent common ancestor, *MY* million years.
[a]Inversions shared with subpopulations other than *GJ* were excluded to avoid counting regions that were possibly introgressed from populations with higher times to MRCA.

within the proto-*japonica* population[38], which can be estimated to be 10,000 generations based on previous estimates of effective population size[37]. For a comparison involving tropical *japonica*, care should be taken to exclude inversions that possibly originated in *Xian/Indica* through introgression. Thus, out of the 431 clustered inversions that were segregated within *GJ*, we focused on 170 that were not segregated within non-*GJ* subpopulations. To deal with the possibility of inversions that originated in *XI* but did not appear in our data set, we also filtered out 26 inverted regions closer to *XI* than to *GJ* based on the 3K-RGP SNP data[39]. The remaining 144 inversions were used to compute the number of inversions between each pair of *GJ* genomes. The average number of (filtered) inversions between two *GJ* genomes is 20.86,

which translates to an inversion rate of 735 inversions per MY. Calculations involving comparisons only between a temperate *GJ* and another *GJ* population lead to a similar estimate of 749 inversions per MY (Table 2 and Fig. 4a).

## Species and population scale analysis

Of the 1426 clustered inversions detected (Supplementary Data 4), we classified them into four different groups: Inversions segregating in *O. sativa* (S), Inversions segregating in both *O. sativa* and *O. rufipogon* (SR), *O. rufipogon* specific (R), and *O. punctata* specific or AA-fixed (i.e., ancestral state not clear) (P) (Fig. 4a). As a result, 1303 (91.4%) appeared to be species-specific, i.e., *O. sativa* (S): 885 (totaling 54.07 Mb),

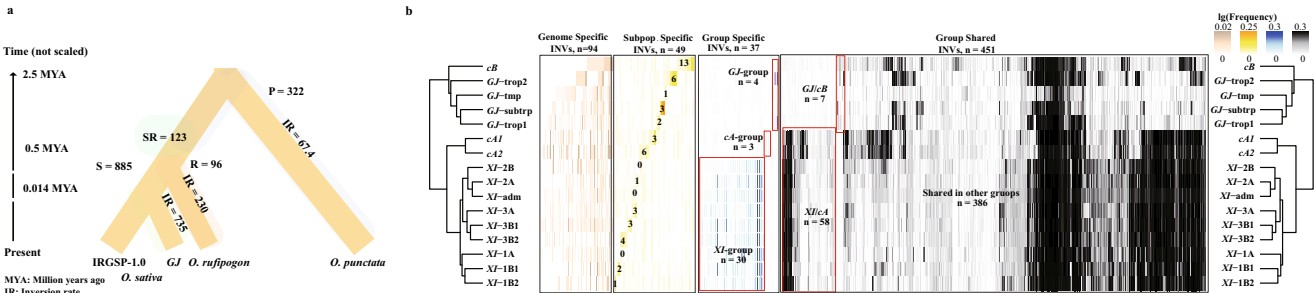

**Fig. 4 | Species-specific, population-specific and shared inversion analysis of the pan-genome inversion index of rice. a** A model showing species specific inversions and inversion rates in Asian rice and two wild relatives (*O. punctata* and *O. rufipogon*). **b** Frequency of genome-specific, subpopulation specific, group specific and group shared inversions across the population structure of Asian rice (*n* = 631). Source data are provided as a Source Data file.

*O. rufipogon* (R): 96 (totaling 4.66 Mb), and *O. punctata* specific or AA-fixed (P): 322 (totaling 10.06 Mb) (Fig. 4a). The *O. sativa* specific inversions (S) were further classified into three categories: i.e., the IRGSP-1.0 RefSeq represents ancestral state (S1), IRGSP-1.0 RefSeq represents the derived state (S2), all *O. sativa* represent the derived state (fixed inversions) (S3), in which 872 (48.7 Mb), 11 (5.2 Mb), and 2 (72.5 Kb) inversions for S1, S2 and S3 were observed, respectively (Supplementary Table 7). The remaining 123 non-specific inversions were found to segregate in both *O. sativa* and *O. rufipogon* (i.e., originated before *O. sativa* and *O. rufipogon* divergence, or introgression) and totaled to about 3.3 Mb in size, including 121 inversions with IRGSP-1.0 RefSeq as the ancestral state (SR1), and 2 inversions with IRGSP-1.0 RefSeq as the derived state (SR2) (Supplementary Table 7).

To gain deeper insight into the distribution of inversions in rice at the population scale, we studied the 872 *O. sativa* specific inversions (IRGSP-1.0 RefSeq has ancestral state) across the 3K-RGP data set[28]. Beginning with a manually curated set of 8720 inversion alignment patterns at their breakpoints (i.e. 872 inversion alignment patterns across ten accessions), we classified their short-read alignment patterns into four categories (i.e., presence of both breakpoints = inversion, absence of both breakpoints = no inversion, presence of a single breakpoint = inversion + deletion, and no data at both breakpoints = NA) (Supplementary Fig. 6a). These categories were then used to train a machine learning workflow to search for the presence or absence of 2,636,928 alignment patterns for 872 inversions across the 3K-RGP data set (Supplementary Fig. 6b, c and Supplementary Table 8). Of note, this analysis identified 241 inversions and 273 accessions with more than 30% missing inversion data and were thus filtered out, leaving a final data set of 631 inversions across 2751 accessions for downstream analysis (Supplementary Data 7 and Supplementary Note 4). These 631 inversions were then classified into genome-specific, subpopulation-specific, group-specific, and group-shared inversions based on their inversion frequencies (Supplementary Data 8). As shown in Fig. 4b, 94 genome-specific, 49 subpopulation-specific, and 37 group-specific inversions (including 30, 3, and 4 specific inversions to *XI*, *cA*, and *GJ* groups, respectively) were observed. We also observed that 451 inversions were shared among different groups, of which 386 (61.1% of total *O. sativa* specific inversions) were shared across the *GJ*, *XI*, *cA*, or *cB* groups, 58 were shared between the *XI* and *cA* groups, and seven were shared between the *GJ* and *cB* groups (Fig. 4b). In total, these results reveal that 91% of the 1426 cluster inversions tested were species-specific (i.e., *O. punctata*, *O. rufipogon*, and *O. sativa*) at the population scale, and may provide clues to their possible roles in speciation over evolutionary time.

### Characterization of transposable element content within inversions and their breakpoints

Transposable elements (TEs) are known to be associated with inversions[11,12]. Thus, we analyzed the TE content across the inversion

index and at their breakpoints. The total amount of TE related sequences present in the pan-genome inversion index was 63.4%, which is significantly higher (Student's test, *p* = 0.0003) than the average TE content present in the 18-data genome data package at 51.3% (Supplementary Data 9). Furthermore, out of the complete set of 1769 inversions, 888 showed partial or total similarity to TEs at both breakpoint regions (+/−100 bp from the breakpoints), and another 389 showed partial or total similarity to TEs in at least one breakpoint. These results demonstrate that TEs are enriched within inversions and their breakpoints in Asian rice.

Analysis of these breakpoints revealed that both long terminal repeat retrotransposons (LTR-RTs, i.e., Ty3-*Gypsy* and Ty1-*Copia*) and DNA TE Mutator-like elements (MULEs) were significantly enriched (student's test, *p* = 5.08E−11 to 0.027), when the frequency of their presence at the 3538 breakpoints was compared to 35,380 randomly selected genomic locations (i.e., ten replicates) (Fig. 5a). We further studied TEs at the breakpoint of each inversion that were shared across all Asian rice genomes. In doing so, we identified 17 TE families (i.e., 13 Ty3-*Gypsy*, 1 Ty1-*Copia*, 2 *CACTA*, and 1 *Mutator*) present at the breakpoints of more than 10 inversions (Fig. 5b and Supplementary Data 10). An example of an inversion enriched in TEs, including the internal and LTR portions of at least three different LTR-RTs, is shown in Fig. 5c.

Since it is known that inverted repeats can trigger ectopic recombination, thereby leading to genome inversions[40,41], we interrogated 100 randomly selected *O. sativa* inversions for direct or inverted repeats in close proximity to inversion breakpoints. This analysis revealed the presence of direct repeats (including microsatellites) at the breakpoints of both ends of 11 inversions, and inverted repeats at single breakpoints of 30 inversions (Supplementary Data 11). For the remaining 118 inversion breakpoints (i.e., 59 inversions), no clear evidence could be found for the presence of inverted repeats at their inversion breakpoints. Of note, our analysis only considered inverted repeats longer than 10 bp, even though shorter have been shown to trigger inversions as well[42]. For this reason, we caution that our estimate of the presence of inverted repeats at inversion breakpoints should be considered on the lower end of the spectrum.

### Characterization of gene content within inversions and their breakpoints

Based on the pan-genome inversion index, we identified a total of ~971 annotated genes per genome within inversions or at their breakpoints (Supplementary Data 12). To investigate the possible effects of inversions on gene expression, we interrogated a set of transcriptome datasets derived from three *O. sativa* subpopulations (i.e., *XI*-adm: MH63, *XI*−1A: ZS97 and *GJ*-temp: Nipponbare (i.e., dataset#2 - see Methods). Based on a comparison of transcript abundance levels between Nipponbare and MH63 and ZS97 (across four tissue types - root, panicle, young leaf, and mature leaf) we detected that 5−12 genes

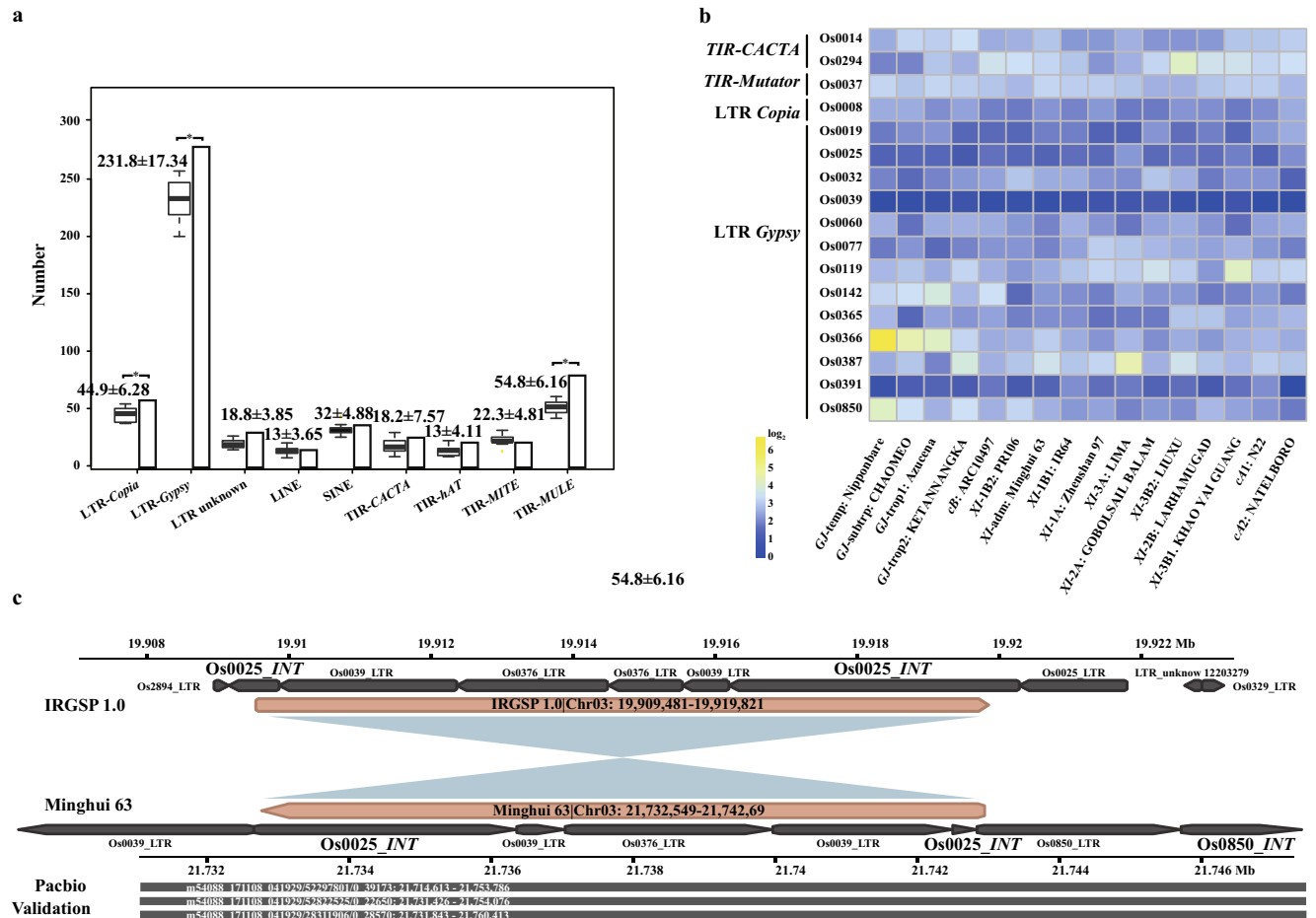

**Fig. 5 | Transposable element analysis of the pan-genome inversions index of rice. a** The amount (y-axis) of different transposable element (TE) families (x-axis) show that three TE families (i.e., LTR-RT Ty1-*copia*, Ty3-*gypsy* and DNA-TE MULE) were observed in higher frequencies at the breakpoints of the pan-genome inversion index than the resampled control tests. Box- and bar-plots show the frequencies of TEs observed at the breakpoints of random resampled regions (*n* = 10 biologically independent resamples) and the pan-genome inversion index, respectively. Each boxplot presents the minimum, first quartile, median, third quartile, and maximum value, and along with mean ± SD (Standard Deviation) are shown. **b** Enrichment/depletion of 17 TEs present at the inversion breakpoints with more than 10 copies. **c** Details of Ty3-*gypsy* Os0025 presence at inversion breakpoints, with support from PacBio long reads. Os0025_LTR (long-terminal repeat), and Os0025_INT (integrase). Source data are provided as a Source Data file.

from MH63 and 4–11 genes from ZS97 within inversions, 2–7 genes from MH63 and 2–4 genes from ZS97 in inverted flanking regions, and 19–42 genes from MH63 and 9–30 genes from ZS97 that were located in non-inverted randomly resampled 20 Kb regions, were differentially expressed (DEG, fold change >2, *p* value <0.01) (Supplementary Fig. 7).

To investigate the effect of inversions on the transcription of genes located at inversion breakpoints - i.e., about 55 genes per genome (Supplementary Data 12), we interrogated both our baseline RNA-Seq datasets (dataset#1- see Methods) and dataset#2 for changes in transcript abundance. On average, transcript evidence for 28 of the 55 genes per genome could be detected in the tissues tested (Supplementary Data 12). Of these, transcript abundance of an average of 20 genes per genome did not change due to the presence of duplicated genes at both ends of their inversion breakpoints (Supplementary Data 12). An example of this observation is represented by two *OsNAS* genes (*NAS1* and *NAS2*) located at the breakpoints of INV0300350 (~4.3 Kb) (Fig. 6a, b). The remaining ~8 genes per genome were single copy and were disrupted the inversion events, leading to the absence of transcript evidence (Supplementary Data 12). For example, transcripts of the Nipponbare *Fbox* gene (Os11g0532600) could be detected in the four tissues tested. However, the first exon of this gene was disrupted in MH63 by INV1101460, resulting in transcript ablation (Fig. 6c, d).

## Recombination rate and genomic inversions

To evaluate the effect of inversions on recombination frequency, a previously published recombinant inbred line (RIL-10) population of 210 inbred lines[43] derived from a cross between *O. sativa cv. XI*-adm: MH63 and *XI*-1A: ZS97 was investigated. We detected 78 inversions between MH63 and ZS97, totaling 3.58 Mb and 3.51 Mb in size, based on the MH63 and ZS97 genome assemblies, respectively (Supplementary Data 13). The recombination rate along each chromosome was assessed by comparing genetic and physical distances between neighboring bins. The average recombination rate for each chromosome ranged from 5.95 (chromosome 6) to 9.92 (chromosome 12) cM/Mb, and varied from 0 to 153.93 cM/Mb across the genome with an average of 6.98 cM/Mb (Supplementary Fig. 8a). The average recombination rate over the 78 inverted regions was 4.00 cM/Mb (0–23.26 cM/Mb), which is significantly lower (Student's *t* test, *p* = 0.0002) than that observed genome-wide (Supplementary Fig. 8b). These results indicate that a marked suppression of genetic recombination is associated with inversions.

## Effect of large inversions on population SNP variation

The occurrence of inversions can affect DNA polymorphism at the population level in several ways, including increased divergence in the inverted region and changes in linkage disequilibrium (LD) patterns[40].

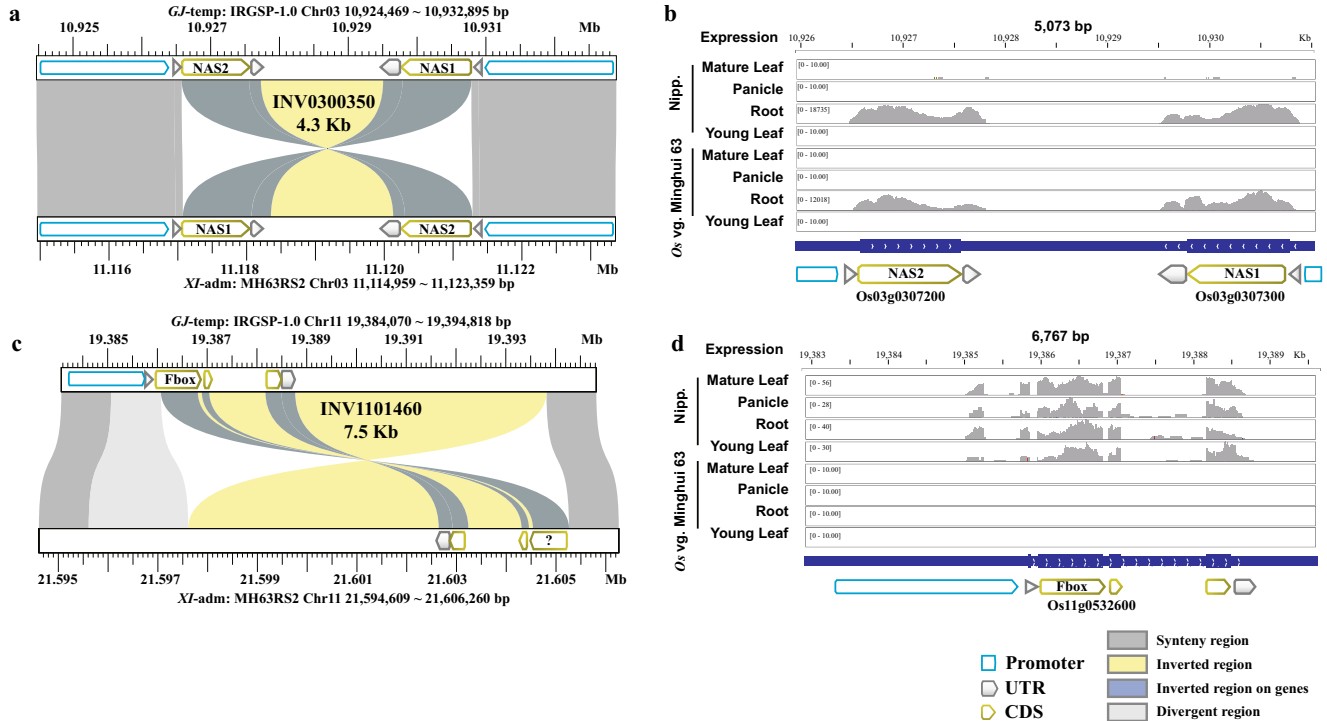

**Fig. 6 | Transcript abundance of genes located at inversion breakpoints. a** Two copies of the *OsNAS* gene lie at the ends of an inversion in the MH63 (*XI*-adm) genome. This inversion disrupted the 5' UTR regions of two *OsNAS* genes (*NAS1* and *NAS2*); **b** *OsNAS* transcript abundance in root tissue; **c** The coding sequence (CDS) of a *Fbox* gene was disrupted by an inversion in the MH63 (*XI*-adm) genome; **d** *Fbox* transcript abundance was suppressed in all tissues tested in the MH63 (*XI*-adm) genome.

The latter is particularly interesting as it can affect SNPs that are mapped to positions megabases apart, and can be a confounding factor in LD-based analyses. To determine whether large *O. sativa* inversions (>100 Kb) left a trace in patterns of LD along the IRGSP RefSeq, we used the 3K-RGP dataset to examine LD blocks near inverted regions (*n* = 88) (Supplementary Data 14). An inversion fixed in a population may lead to the disruption of LD blocks, in which some SNPs flanking the inversion on one side are in LD with SNPs on the distal part of the inversion, but not on the adjacent part (Fig. 7), due to the reversed order of SNPs inside the inverted region in samples that carry the inversion allele. By an LD block we mean only a set of SNPs in high LD ($r^2$ > 0.8 in this analysis).

Next, we examined the entire 3K-RGP variation data set and searched for LD blocks that connect the flanking regions of inversions, having no SNPs in the proximal parts of each inversion. Such blocks (Fig. 7) were found in nearly all large inversions (126 out of 147 [85.7%], or 74 out of 88 [84%] inversion clusters) (Supplementary Data 14) with exceptions falling into two categories: i.e., inversions in regions of complex chromosomal rearrangements (Chr04:14.1–15 Mb, Chr11:6.6–6.85 Mb, Chr11:9.4–9.7 Mb), and three putative recent inversions (INV0401400, INV0500310, INV1000040), each of which were found in single genomes and may lack sufficient frequencies in a population to contain traces of recombination. Some of the disrupted LD blocks contained a particularly large number of SNPs and were seen as a distinctive checkered pattern on LD heatmaps (Fig. 7). This comparatively large number of SNPs along with low haplotype diversity, despite the presence of recombination, could be a consequence of selective pressure.

## Discussion

Inversions are an important class of structural variations that have been shown to play important roles in the suppression of recombination that can lead to the selection of adaptive traits, reproductive

isolation and eventual speciation, and are quite common in plants[31,40,41]. For example, over the 50–60 MY history of the *Poaceae*, where gene order has been largely conserved, Ahn and Tanksley showed (using molecular genetic maps) that multiple inversions and translocations occurred during the evolution of maize and rice from a common ancestor[42].

Here, we present a comprehensive analysis of the inversion landscape of Asian rice at the population structure level with the discovery of 1769 non-redundant inversions that range in cumulative size from 0.3 Mb to 15 Mb (Supplementary Data 3). This analysis allowed us to estimate inversion rates at multiples levels (i.e., genome type (AA vs. BB: 67.4 inversions per MY), AA species (*O. sativa* vs. *O. rufipogon*: 320 inversions per MY), and *O. sativa* populations (K = 15: 735–749 inversions per MY), all of which were 2.1–4.3×, 10.6–21.3×, and 16.6–50× higher than previously estimated[40], respectively. Huang and Rieseberg[40] predicted that these earlier inversion rate estimates for plants should be considered as underestimates (15–30 inversions per MY), as they were dependent on the quality of the genomes analyzed, and other factors, which bears true with our analyses. These inversion rate estimates over such a short time period may be reflective of high fixation rates of rearrangements in plants[44], high chromosomal evolution rates in annual plants[45–47], and intense human selection since the dawn of agriculture[11,28].

Pan-genomes, first described in bacteria[48], can be defined as a representation of the entire set of genes within a species, consisting of a core genome - containing sequences shared between all individuals of the species - and the 'dispensable' genome. To date, five pan-genomes have been published for Asian rice (summarized in Table 3), where three reported inversions, none of which were validated. To compare our data with the first reported inversion genome scan[11], we extracted the inversion coordinates (i.e., 2402 INVs, average total length 43.3 Kb) from the same set of 15 accessions used to generate our Rice Population Reference Panel and found that only 200 could be

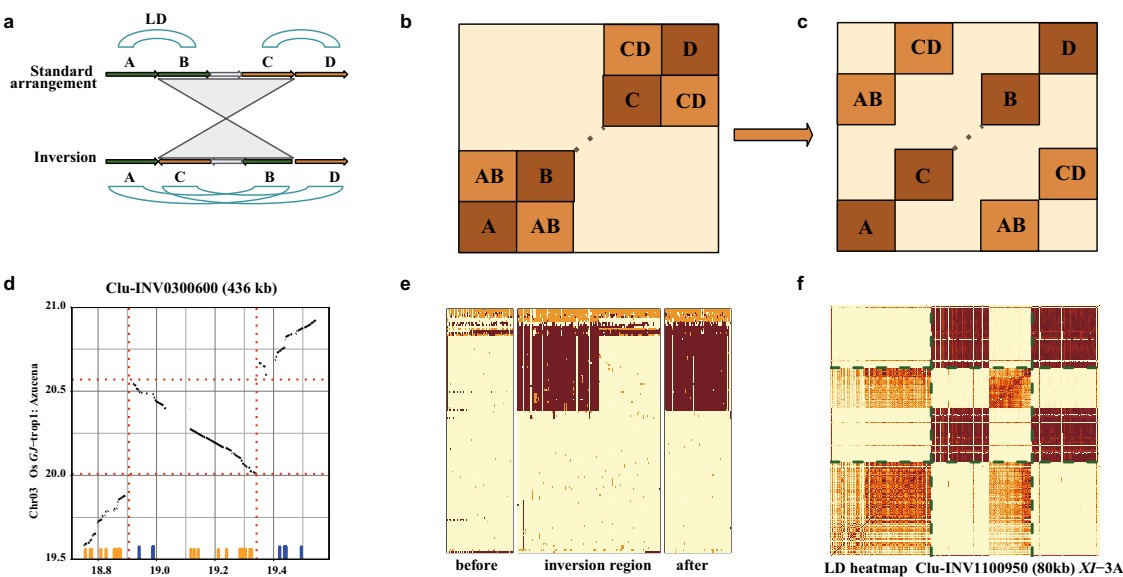

**Fig. 7 | Population linkage disequilibrium analysis of large inversions.** A schematic diagram of linkage disequilibrium (LD) block disruption arising from the presence of an inversion, as shown in A and B. **a** Cartoon view of an inversion with breakpoints disrupting two LD blocks; **b** Expected features of the corresponding LD heat map; **c** Example of SNP blocks in high LD that are disrupted by an inversion; **d** The panel shows alignments, with the inversion marked by dotted lines. Small vertical lines above the horizontal axis mark the location of SNPs constituting a disrupted LD block. Orange and blue colors delineate two LD blocks that are contiguous in the of *GJ*-trop1 population, but appear as split when aligned to the IRGSP RefSeq (*GJ*-temp). Disruption of Azucena (*GJ*-trop1) haplotype blocks along the IRGSP RefSeq in the region of INV030410, as shown in **e** and **f**; **e** Genotype heat map of the *GJ*-trop1 subpopulation (samples in rows, SNPs in columns; light yellow: reference call, orange: heterozygous, brown: homozygous variant); **f** LD heat map of the same subpopulation. Dotted lines show the inversion region. Darker colors show larger $r^2$. Note that the scaling of X-axis in the genotype heat map is not uniform, allotting half of X-axis space to the inverted region. Source data are provided as a Source Data file.

validated with dot plots (a 91.7% false positive rate) (Supplementary Table 9), 196 of which overlapped with our inversion index. The 6 remaining contained 2 that overlapped, and only 4 that were not present in our inversion index (Supplementary Data 15). These analyses combined reveal the limitations of inversion callers with short read data and provide a cautionary note as to the validity of many of the inversions cataloged to date.

More recently, Qin et al. identified 718 inversions using 29 PacBio genomes of Asian rice mapped to the IRGSP RefSeq, 557 of which overlapped with our analysis and the remaining 161 could not be validated. Although informative, a combination of genomes that bridge the $K = 15$ population structure of Asian rice and a minimum of 60 genomes are required for a comprehensive analysis of inversions in Asian rice, as demonstrated here (Fig. 1).

Several key factors led to our ability to generate a definitive pan-genome inversion index for cultivated Asian rice. The first was our use of a set 75 high-quality reference genomes that span the $K = 15$ population structure of Asian rice[32] (Supplementary Data 2), plus 2 high-quality genomes from the wild ancestors of rice that served as phylogenetically anchored outgroup species (Supplementary Table 1). Secondly, we did not computationally collapse these 75 genomes into a pan-genome (e.g., genome graph), but maintained all 75 rice genomes in their native state. This was key to our ability to precisely compare all genomes one-by-one. Lastly, we interrogated the 3K-RGP data to estimate and validate the population genetics of each inversion. As sequencing costs continue to plummet, the ease at which ultra-high-quality genomes can be generated, and with computational power exceeding current limits[49–51], we predict that there will no longer be a need to computationally generate pan-genomes to perform similar analyses as demonstrated here across much larger genomes, such as wheat (genome size = 15 Gb)[52–54].

The Asian rice pan-genome inversion index is the first step on our quest to precisely discover all standing natural variation that exists in Asian rice and eventually the genus *Oryza* as a whole. The next step will

be the generation of a digital genebank for Asian rice whereby resequencing data from >100,000 accessions will be mapped to our Rice Population Reference Panel. Preliminary data (unpublished) shows that we can now easily call SNPs with resequencing data from >3000 individuals in 5 days per genome or less using high-performance computational workflows optimized for GATK4[55] software. Such call rates will undoubtedly increase over the next year with a targeted rice digital genebank release date of January 1st, 2025.

## Methods

### Sequence and assembly

Seed and/or tissue from *O. rufipogon* (IRGC 106523) and *O. punctata* (IRGC 105690) species were obtained from the International Rice Research Institute (IRRI), Philippines. The *O. rufipogon* accession was selected as a true *O. rufipogon* representative based on its origin (Papua New Guinea, reproductively isolated from rice cultivation) and phenotype. The two genomes were sequenced to a minimum of 100× coverage using PacBio long-read technology (PacBio RSII), and were assembled and validated to a PSRefSeq quality level following the identical strategy as described in Supplementary Note 1. The Benchmarking Universal Single-Copy Orthologs (BUSCO v4.0) software package[56] was employed to evaluate the gene space completeness of each assembly.

### Genome annotation

Genome annotation used PacBio Iso-Seq and Illumina RNA-Seq data derived from RNA isolated root, panicle, and young leaf tissue from Nipponbare and 13 newly sequenced *O. sativa* accessions[32] as baseline transcript evidence (i.e., RNA-Seq dataset#1). In addition, we collected deep RNA-Seq data from *O. sativa* cvs. Nipponbare, Minghui 63 and Zhenshan 97 (RNA-Seq dataset#2[57]) for upgrading their annotations with the same pipeline and downstream transcriptome analyses. PacBio Iso-Seq data was deposited in NCBI under BioProject PRJNA760839. The RNA-Seq dataset#1 is deposited in NCBI under

**Table 3 | A comparison of public pan-genome research in Asian rice**

| Pan-genome studies | #genomes | Species | Contig N50s | Assembly | Subpopulations (K) | Sequencing Technology | Sequence validation | Inversion callers | Validated | # INVs | Data access |
|---|---|---|---|---|---|---|---|---|---|---|---|
| Wang et al.[28] Fuentes et al.[11] | 3K | O. sativa | 5.44 ± 2.87 Kb | Contigs/ scaffolds | 9 | Illumina | NA | DELLY & LUMPY | NO | 210,301 (raw) | https://www.ncbi.nlm.nih.gov/sra/?term=PRJEB6180 https://snp-seek.irri.org/ |
| Zhao et al.[92] | 66 | O. sativa, O. rufipogon | 40.5 ± 13 Kb | Contigs | 4 | Illumina | NA | NA | NA | NA | http://www.ncgr.ac.cn/RicePanGenome |
| Qin et al.[12] | 33 | O. sativa, O. glaberrima | 12.6 ± 4.8 Mb | Chromosome | 9 | PacBio | BioNanao HiC | Nucmer & SyRI | NO | 954 | http://www.ricerc.com/ |
| Zhang et al.[93] | 111 | O. sativa, O. rufipogon | 7.4 ± 2.1Mb | Chromosome/ scaffolds | 5 & 9 | PacBio & ONT | NA | NA | NA | NA | https://cgm.sjtu.edu.cn/TGSrice/index.html |
| Shang et al.[94] | 251 | O. sativa, O. rufipogon, O. glaberrima, O. barthii | 10.9 ± 3.7 Mb | Contigs | 2/15(174 accessions from 15 subpopulations of 3K-RGP, GJ-trop2 is missing) | ONT | HiC | Minimap2, NGMLR, and Sniffles | NO | 2784 (raw) | http://www.ricesuperpir.com/ |
| Our study | 75 | O. sativa, O. rufipogon, O. punctata | 22.7 ± 8.2 Mb | Chromosome | 15 | PacBio & ONT | BioNanao | Nucmer & SyRI | YES | 1769 (validated) | https://yongzhou2019.github.io/Rice-Population-Reference-Panel/ |

BioProject PRJNA659864. RNA-Seq dataset#2 is retrieved from NCBI BioProject PRJNA597070[57].

Protein coding genes for the 16 *O. sativa* genomes above were predicted using MAKER-P (3.01.03)[58] including expression evidence, homology and ab initio gene predictors FGENESH (v.8.0.0)[59,60], SNAP (0.15.7)[59,60]. Repeat masking was performed with RepeatMasker (http://www.repeatmasker.org) using an *Oryza* specific repeat library[31]. Expression evidence included reference guided transcript assemblies generated using StringTie (v1.3.4a)[61] and Cufflinks (v2.2.1)[61,62]. To generate assembled transcripts, quality inspected RNA-Seq reads from each library were mapped to their respective genomes using STAR (v2.5.3a)[63] with an iterative 2-pass mapping approach in which splice junctions generated from the first round were used to refine alignments in the subsequent round. Mapped reads from each library were merged, sorted, and indexed using SAMTools (v1.9)[64] to generate input for transcript assembly programs. All software packages were run with default options. High quality full length transcripts were clustered using CD-HIT (v4.6)[65] for 95% sequence identity using parameters -c 0.95 -n 10 -d 0 -M 3000. The clustered transcripts were further filtered for intron retention events using SUPPA2[66]. Additional transcript and homology evidence was used[31] to run MAKER-P with keep_preds option set to 1. The gene structure of the predicted models was further improved using PASA (v2.4.1)[67] using full length cDNA and EST's downloaded from GenBank with the query "EST [Keyword] AND *Oryza sativa* [Organism]". Functional domain identification was completed with InterProScan (v5.38-76.0)[68]. TRaCE (https://github.com/warelab/TRaCE)[69] was used to assign canonical transcripts based on domain coverage, protein length, and similarity to transcripts assembled by Stringtie (v1.3.4a)[61]. Annotation quality was assessed with MAKER-P generated Annotation Edit Distance (AED) values[70] and BUSCO, respectively. Only transcripts with AED scores <1 were retained. Finally, gene annotations were imported to Ensembl core databases, verified, and validated for translation using the Ensembl API[71]. All genome annotations are available at the Gramene Pan *Oryza* database (https://oryza.gramene.org/) and the Rice Population Reference Panel (RPRP, https://yongzhou2019.github.io/Rice-Population-Reference-Panel/data/).

### Genome visualization

For alignment, analysis, visualization, and public availability, we uploaded the 18-genome data package into the Persephone® multi-genome browser (https://web.persephonesoft.com/). The types of data tracks visualized include gene models, marker locations, BLAST matches, RNA-Seq coverage and sequence tracks. As the genomes are closely related, the maps could be aligned by connecting short sequence tags (100 bp) derived from the IRGSP-1.0 RefSeq and mapped onto the other 17 genomes using BLASTN, as shown in Supplementary Fig. 2.

### Transposable element annotation

All 16 cultivated *O. sativa*, plus the *O. rufipogon* and *O. punctata* genomes were re-annotated using the output of the latest version of the EDTA (v1.7.4)[72] transposable element (TE) annotation pipeline. The entire output was loaded into RepeatMasker (v 4.0.8)[73], with the exception of predicted helitron elements that were skipped because of a high false positive rate.

### Identification of genomic inversions and assessments

To discover large inversions (>100 bp), we first tested four different analysis workflows (https://gitlab.kaust.edu.sa/zhouy0e/sv-for-o.sativa)[74] on two genomes, i.e., GJ-temp: IRGSP-1.0 and *XI*-adm: MH63 (the genome of Minghui 63 accession).

Workflow 1: The MH63 genome assembly was split into overlapping reads of 50 Kb in length at 5 Kb step intervals, resulting in ~10× coverage. The reads were then mapped onto the IRGSP-1.0 genome

sequence using the tool CoNvex Gap-cost alignMents for Long Reads (NGMLR, v0.2.7)[75]. Inversions were called with SVIM (v1.1.0)[76], retaining only inversions with a depth greater > 6 that passed the caller's filtration criteria.

Workflow 2: Steps were the same as in workflow 1, except that Sniffles (v1.0.7)[75] was used to call inversions.

Workflow 3: The MH63 genome assembly was aligned to IRGSP-1.0 reference sequence using Minimap2 (v 2.15)[77]. This step was followed by filtration for identity greater than 90% and length longer than 100 bp. Inversions were called using SyRI (i.e., Synteny and Rearrangement Identifier, v1.4)[78].

Workflow 4: Steps were the same as in workflow3, except the alignment tool used, i.e., Nucmer, included with the MUMmer (v4)[79] software package.

To assess the accuracy of the four workflows, dot-plots of the syntenic regions including the putative inversions were generated to visually validate the inversions. Out of all workflows, workflow4 was selected based on our validation criteria. As a final validation check of workflow 4, PacBio raw reads were also applied for validation of a subset inversions. Raw reads of query genomes were mapped to the IRGSP RefSeq, and the assemblies of the query genomes themselves. If breakpoints could be observed in the reference but not in the assemblies of the query genomes, the breakpoints were called as 'supported' by the PacBio raw reads, and vice versa.

Subsequently, the sequences of all 74 genomes were aligned to the IRGSP RefSeq with the MUMmer[79], filtering the alignments for a minimum identity of 90%, and minimum length of 100 bp, and the coordinates were retrieved using the function "show-coords" of MUMmer[79]. Finally, inversions were called using the SyRI tool (v1.4)[78] with default parameters, which provided VCFs (v.3) with ID, start, end of reference and query genome coordinates that was leveraged for pan-genome comparisons downstream.

### Pan-genome inversion index of 75 high-quality genomes

Seventy-four inversion vcf files derived from 74 genomes compared to the IRGSP RefSeq were generated. Briefly, vcf files were sorted according to chromosome ID, coordinates and strand, and were merged using SURVIVOR (v1.0.7)[80], run under default parameters (https://github.com/fritzsedlazeck/SURVIVOR/wiki), except for the maximum allowed distance of 10 bp. In this case, inversions having start and end coordinates no more different than 10 bp were collapsed and considered as single inversions.

### Genome-wide distribution of the pan-genome inversion index

To investigate the genome-wide distribution of the pan-genome inversion index the Kolmogorov–Smirnov (KS) test was performed on the coordinates of inversions[81,82]. For each chromosome, we performed 10,000 simulations of uniformly distributed positions of the same number as the inversions reported for each chromosome and calculated $p$ values (the rate of uniform inversion distribution in simulations ($p$ values > 0.05) based on the Monte Carlo method. In addition, inversions were further classified into different lengths (<1 Kb, 1–5 Kb, 5–10 Kb, and >10 Kb) and tested for uniform distribution independently. The $p$ values were adjusted using the Benjamini–Hochberg procedure[83,84] to reduce the false positive rate on a per-chromosome basis.

### Inversion rate estimations

To estimate the inversion rate (IR) across the pan-genome of Asian rice, we considered pairs of populations or genomes with existing estimates of divergence times to a most recent common ancestor (MRCA), and divided the total number of inversions by twice the time to the MRCA (TMRCA, corresponding to the total branch length of the genealogy on two nodes), i.e. the estimate is calculated using the following equation:

$$IR = \frac{Number\ of\ INVs}{2 \times TMRCA} \quad (1)$$

### Differential transcript abundance analysis within inversions

To investigate the effect of an inversion on gene expression, we searched for differences in transcript abundance within inversions in Asian rice based on two RNA-Seq datasets (dataset#1 and dataset#2). FPKM (Fragments Per Kilobase of exon model per Million mapped fragments) values were identified following an accurate pipeline[85], which included HISAT2 (version 2.2.1)[86] for alignment, and StringTie (v1.3.4a)[61] to obtain normalized FPKM values. Taking into account different transcript abundances between the two RNA-Seq datasets (2 Gb for dataset#1 vs 6 Gb for dataset#2), we used different minimum filtration of, i.e., FPKM > 0.1 in RNA-Seq dataset#1 and FPKM > 1 in RNA-Seq dataset#2. Differential Transcript Abundance (DTA) of dataset#2, was carried out by edgeR (3.1.0)[87] with $p$ value <0.01 and abundance change >2 cutoffs.

### Genome recombination rate of inversions

To study the genome recombination rate of inversions, we used genetic data based on a RIL population derived from accessions *O. sativa* cv. *XI*-adm: Minghui 63 and *XI*−1A: Zhenshan 97[88,89]. Briefly, re-sequencing data was obtained from 210 RILs and a Bin Map containing 1619 bins was generated. The physical positions of each bin were derived from updated versions both the MH63 and ZS97 reference genomes[89]. Then, a genetic map based on the RILs panel of 1619 bins was constructed using the MSTMap algorithm[90]. By comparing genetic and physical distances between neighboring Bin Map markers, we estimated the relative changes of the recombination rate of the two genomes[91]. Then, we compared recombination rates in the bins that overlapped with inversions and genome-wide, respectively.

### Reporting summary

Further information on research design is available in the Nature Portfolio Reporting Summary linked to this article.

## Data availability

All sequencing data has been submitted to NCBI. The assembly data of IRGSP-1.0/NIPPONBARE (GCA_001433935.1), CHAO MEO::IRGC 80273-1 (GCA_009831315.1), Azucena (GCA_009830595.1), KETAN NANG-KA::IRGC 19961-2 (GCA_009831275.1), ARC 10497::IRGC 12485-1 (GCA_009831255.1), PR 106::IRGC 53418-1 (GCA_009831045.1), Minghui 63 (GCA_001623365.2), IR 64 (GCA_009914875.1), Zhenshan 97 (GCA_001623345.2), LIMA::IRGC 81487-1 (GCA_009829395.1), KHAO YAI GUANG::IRGC 65972-1 (GCA_009831295.1), GOBOL SAIL (BALA-M)::IRGC 26624-2 (GCA_009831025.1), LIU XU::IRGC 109232-1 (GCA_009829375.1), LARHA MUGAD::IRGC 52339-1 (GCA_009831355.1), N22 (N 22::IRGC 19379-1) (GCA_001952365.2), NATEL BORO::IRGC 34749-1 (GCA_009831335.1), *O. rufipogon* PNG91-7::IRGC 106523-1 (GCA_023541355.1), and *O. punctata*:: IRGC 105690 (GCA_000573905.2) were deposited in GeneBank of NCBI. The raw sequencing data of *O. rufipogon* and *O. punctata* generated in this study were deposited in NCBI under BioProject PRJNA609053 and PRJNA13770, respectively. PacBio Iso-Seq data for all accessions in this study was deposited in NCBI under BioProject PRJNA760839. The RNA-Seq dataset#1 was deposited in NCBI under BioProject PRJNA659864. RNA-Seq dataset#2 was retrieved from NCBI BioProject PRJNA597070. The details are also listed in Table 1 and the Rice Population Reference Panel [https://yongzhou2019.github.io/Rice-Population-Reference-Panel/data/]. Source data are provided with this paper.

## Code availability

Code for the different inversion calling workflows are available at the Rice Population Reference Panel [https://yongzhou2019.github.io/Rice-Population-Reference-Panel/software/][74].

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

## Acknowledgements

This research was supported by King Abdullah University of Science & Technology's Baseline funding, and the University of Arizona's Bud Antle Endowed Chair for Excellent in Agriculture to R.A.W.; Huazhong Agricultural University's Start-up Fund, Fundamental Research Funds for the Central Universities (2662020SKPY010), and the Major Project of Hubei Hongshan Laboratory (2022HSZD031) to J.Z.; USDA ARS 8062- 21000-041-00D to D.W. The authors acknowledge support from the Shaheen Cray XC40 platform at KAUST Supercomputing Laboratory, the computing platform of the National Key Laboratory of Crop Genetic Improvement at HZAU, Peter VanBuren for systems support and the Elzar High Performance Computing facility (NIH S10 OD0286321-01) at Cold Spring Harbor Laboratory for providing computational resources, and help from the Persephone team that hosts our 18-genome data package on their portal for genome visualization.

## Author contributions

A.Z., D.W., K.L.M., J.Z., and R.A.W. designed and conceived the research. K.M. and IRRI provided seed and/or tissue for all *Oryza* accessions. D.K., N.M., D.Ch., and M.L. performed DNA extractions and genome sequencing. Y.Z., Z.Y., and J.Z. performed sequence assembly, GPM edit and validation of 18 genome sequences. Y.Z., N.M., D.K. and V.L. carried out the optical map sequence and analysis. D.Ch., Y.Z., J.S. and K.L.M. performed population genetic analysis. K.C., Z.L., and D.W. performed the genome annotation and validation. Y.Z., Z.Y., J.Z. and A.Z. performed the gene expression analysis and TE annotation. Y.Z., A.Z., J.S., A.A., S.M., Z.Y., and J.Z. carried out the inversion identification and population level validation. M.Th., L.R. and Y.Z. performed the machine learning study for inversion detection. L.R., N.K., M.Tr., C.D.G., and K.C. managed the computing platforms. Y.Z., Z.Y., D.Co., K.C., N.A., A.F., A.Z., K.L.M., J.Z., and R.A.W. wrote and edited the paper. All authors read and approved the final manuscript.

## Competing interests

The authors declare no competing interests.
