## [Peer Review File · Nature Communications]

Pan-genome inversion index reveals evolutionary insights into the subpopulation structure of Asian rice (*Oryza sativa*)Reviewers' Comments:

Reviewer #1:

Remarks to the Author:

This manuscript presented 16 rice high quality assemblies, including cultivated and two wild species, and focused on the analysis regarding inversions in terms of evolution, effects on gene regulation, recombination rate et al. I have several major concerns as follow:

1: The authors have tried to resolve inversions in population scale by combining 18 rice high-quality assemblies. However, to my knowledge, up to 60 rice high quality assemblies have been available so far, it's much better to combine all available assemblies together to investigate inversions at population level. Moreover, the authors investigated inversion distribution in 5 additional wild species with high-quality genomes (line 246-250), unfortunately, they were noted "unpublished data", those wild species will enhance the quality of this study.

2: The authors should compare the inversions in this study with inversions which had been published, such as that data in Qin et al. Cell, 2021, wherein they also identified tons of inversions. Because the authors did not perform the comparison, so the author can determine how many new inversions have been identified, and therefore, the description at line 236 is not accurate: "two of which (INV060390 and INV080710) were previously reported". Moreover, the descriptions in the section of "five largest inversions" are boring, it was just a list of the distribution of these inversion, and some distribution information have been revealed before.

3: In the next four sections after "five largest inversions", I did not see any attractive and new contents which will significantly promote rice biological and evolutionary studies. For example: the conclusion in the section "Characterization of Transposable Element Content within Inversions and Breakpoints" (our results reveal an enrichment of TE related sequences both within inversions and at their breakpoints), have been reported in several papers. Same thing for the conclusion "a marked suppression of genetic recombination is associated with inversions" of the section "Recombination Rate and Genomic Inversions".

The investigation about the effect of inversions on the genes located within inverted regions, and their expression wasn't appropriate, the effect of other sequence variations, such as SNP and SV, were not considered, these variations were supposed to have much bigger and direct impacts on gene expression than inversions, whereas, they were overlooked in this investigation, and without discussions. Additionally, regarding the section "Phenotypic consequences of inversions: Inversion Cluster 92", based on the description and my understanding, it was much better to say the consequence of SNP.

Reviewer #2:

Remarks to the Author:

This paper details the comparison of 18 long-read based genomes of rice. Of the 18, 16 had been published previously, one was updated and one was new. The goal of the study was to use high-quality genomes to infer the position and prevalence of inversions, which are likely to be an important class of structural variant (SV). There have been many studies of rice genomes recently, so the novel contribution of this one is its focus on inversions events. As set forth at the end of the Introduction, the paper makes five main claims – i.e., a catalog of 1054 inversions, an inversion rate, and some biological effects (lower LD and at least one inversion that contributes to delayed flowering).

I have many comments on the paper, and list them according to my progress through the paper. Many are admittedly quite minor and are offered in the spirit of helpful criticism. However, others are, I believe, more substantive and require substantial revision and perhaps more thought. Overall, the

data presented in the paper did not, in my mind, match the claims of the paper.

- Given that inversions are the main thrust of the paper, the paper is narrowly written, with a focus only on rice. There is lots of previous evidence – although mostly not discussed in this manuscript – that inversions affect phenotypes, gene expression, are mediated TEs, etc. From my perspective, it's a missed opportunity to not put this rice work into the context of the wider plant literature (maize, tomato, grapes, evolution, etc. etc.). The only real attempt at generality is lines 90 to 93, and a bit in the first paragraph of the Discussion.

- The new or updated genomes from rufi and punct are diploid, I think – i.e., from not a naturally selfing lineage, like rice. How were diploid genomes treated? Details on phasing, haplotypes and heterozygosity are lacking, but probably important (particularly if diploid chromosomes have SVs, so that their treatment is key to inferences)

- This is admittedly a stylistic thing, but I find it awkward to list the main results at the end of the Discussion. I'd rather know what questions and going to be asked and why. Moreover, on careful reading, I feel as if at least #2 and #3 were overstated.

- I struggled to follow the sampling and the nomenclature (e.g., Xi-adm MH63). It'd be nice to have a sampling table with acronyms, taxon of origin, etc.

- I'm confused by a great deal about the pan-genome (1st two sections of results), because I am not sure how genomes were combined to create a pan-genome. And later, the paper extols the virtue of using a pan-genome free approach. I think (?) that the term pan-genome was used mostly to say "we looked at all of the genomes", but if so that usage is a bit confusing given that pan-genome has gained a more specific meaning.

- p. 175 – I appreciate the detail given to the various workflows in the M&M to estimate pairwise inversions. It gives the impression of great care!

- Line 181 – is this statement relative to IRGSP or to Oruf and Opunct? Generally lines 181 to 190 were pretty tough to follow, given lack of knowledge about sampled genomes. Again, a table would be nice or more explanation.

- Line 200 – I desperately would have liked to see the information in this paragraph summarized on a phylogeny, showing species and group-specific number of inversions on nodes. Given that this paper repeatedly touts a strong phylogenetic basis as the rationale for sampling, the lack of a phylogeny is a somewhat glaring omission.

- I was a bit confused by Figure 1. It's nice in the sense that it shows all of the data, but I think there are no species-specific inversions in rice (vs. other species), right? And what does each blue bar represent? I assume each one is an inversion. It seems to me that there are many inversions shared among the Asian Rice group XI on the right hand side of the graph, but not circled as group specific. Perhaps it's a question of the level of detail, because it is very hard to show all of the inversions, but unfortunately I did not find the figure particularly useful. (I'd love to see inversions in the context of chromosomes, centromeres, etc.)

- Line 208 – are subpopulations groups? (as in Figure 1?)

- Line 216 – if I followed correctly, the 3K-RGP is a short-read dataset. It's not clear to me how that could be used to test/confirm inversions, particularly since later the claim is made that short read data are not useful for inferring inversions after comparing the results of this paper to the Fuertes paper. Moreover, details of methods, numbers used, 'high coverage', etc., are lacking, such that is hard to follow the basis for conclusions.

- Line 256 - TE information about breakpoints is interesting. By "analyzing TE content across the inversion index", is this all 1054 inversions?

- Figure 3C. I'm confused and need some explanation in the legend. The lines in the middle seem to be inverted, but the arrows go in the same direction. The TE (0025) seems to be an LTR that is split in both cases. What is INT?

- How many inversions were validated with bioNano, as in Figure 2? Overall, I'm not convinced that all 1054 inversions were independently validated (as claimed in point #2 in the Intro). I do believe bioNano, but details are lacking (how low is the resolution? How many could be confirmed?) I believe that short-reads are useful, but details are lacking here, too, but (again) its hard to claim they validate inversions on the one hand but are not useful at all on the other (e.g., Fuertes).

- Lines 285 - I'm assuming that 10.9% and 7.3% is much higher than the genome average, but it'd be nice to have an explicit comparison to the genome average to drive this point home.

- About the inversion rate estimates in the Discussion (line 411 and following). It may be that I'm misreading things, but I think they are generally wrong and perhaps horribly so. Here's why. There may be 22 post-inversion events in rice, but (if one thinks in phylogenetic terms) there are many many more years accumulated across the rice lineages on the tree than 10,000. As a brief example, let's assume (for simplicity) that 17 rice genomes diverged 10,000 years ago. If that were true, then the numerator in the rate calculation should be $17 \times 10,000$ years, not 10,000 years, so that estimate is inflated about 17-fold. Of course, we don't know exactly when each of the separate rice genomes diverged from one another, so the estimate of $17 \times 10,000$ years is too many. But hopefully the point is made that the calculations reported in this section may be way off and that the problem may require some consideration of population genetics given the sample.

- Line 432 - the comparison to Fuertes. I'd have to read that paper carefully to see how it was done, but the title implies there were 3,000 individuals. That suggest, I think, that the entire dataset was used as evidence to support or not the inference of inversions. I'd expect them to have found many more inversions, but that the inversions in this paper would be a subset of their total set. I think that'd be a more fair comparison. That said, it is indeed puzzling that there is only 194 out of 1054 that overlapped! While I agree with the authors that short read data is certainly less accurate than long-read data, it does make one wonder about the accuracy of the 18 assemblies and whether there is no only errors in Fuertes but also in assemblies that mislead inversion inference.

- One time consuming but convincing way to validate inversions is to find long reads that span the inferred junctions. I don't think that was done her, but it would certainly go a long way to confirming the inversion inferences more convincingly.

Reviewer #3:

Remarks to the Author:

Zhou et al. present a pan-genome analysis of the major sub-populations of asian rice and two wild rice species represented by a set of 18 whole genome assemblies. the two wild rice assemblies are new and are provided with this manuscript. The other assemblies have been published before. The authors main point of analysis is the cataloging of inversions larger than 100 bp and contextualizing these with data on recombination, LD, selection and gene expression. As the authors point out, the detection of inversions (also larger) is not novel for plant genomes (maybe more references to recent pan-genome studies in cereals would be justified here?), however, here maybe a first comprehensive catalog for representatives of a crop species' subpopulations is provided.

The study reads well and is well presented. The data displays sometimes are rather basic - often just direct exports out of commercial analysis and visualization software? Intuitivity of the figure displays could be improved - maybe?

The study addresses an obvious point in comparative genomics as more genomes of the same species become available. The authors are well aware of artifacts introduced into analyses when different assembly qualities affect the analysed assemblies. Therefore, I was surprised, given the today's costs for making a HiFi assembly for rice (haploid assembly consumable costs in the few hundred dollars range!) that the authors did not make an effort here to have really absolutely comparable datasets. This may only incrementally change the presented results, however, it is a weakness of the study that could have been avoided with modest effort and investment. The same argument applies to the annotation which was done with the same pipeline only for the Asian rice.

The spectrum of inversion distribution: the authors say they are randomly distributed - Ext. figure 3 should support this claim. While the figure legend is not conclusive and the figure itself is basic, I think the authors should have made an effort define size classes and redo the genome distribution scan. Is it true that all sizes of inversions are evenly distributed along the chromosomes? This is counterintuitive.

The authors used the 3K rice data to genotype for the presence of inversions, however, they performed this only on a selected set of genotypes representing the 15 sub-populations. Is it really computationally so intense to include all 3000? Or is this a problem of sequence coverage in a certain proportion of the 3K dataset? I would love to see the analysis on all 3000! I wonder whether the authors tried to use the inversion catalog as a proxy to model and detect additional inversions in the full 3K dataset as the 15 genotypes sequenced will for sure not give the full pan-genome inversion spectrum. The authors outline the technical feasibility in the discussion. Furthermore, it would be very instructive to give an estimate for WGS coverage to robustly scan for the presence of inversions at population scale?

TE landscape was analysed in context of inversions and it was detected that inversions are enriched with TE content in rice. The analysis of inversions could go further here. Have you systematically assessed the TE at the Inversion junctions? This could reveal patterns of the mechanistic involvement for the occurrence of inversions in the rice genome.

Gene expression and inversion: the authors showed examples where inversions and expression levels of genes in inverted regions and their orthologs in "non-inverted" haplotypes. The data is interpreted as inversions are causal, which is not unlikely, however, the authors do not show any functional validation (which is not trivial) and analysis and they also do not provide complementary datasets (methylation, HiC, ATAC etc.) which would support their hypothesis and interpretations. I would also recommend to expand the analysis to genes adjacent or in neighborhood but not directly affected by the inversions - regulatory sequence context can and will be affected also for such genes and one should see a gradient perhaps?

Similarly, the interpretation of positive selection in inverted or non-inverted haplotypes should maybe presented and discussed with more caution.

The authors make a point in the discussion that not merging sequences into a pan-genome graph was an advantage here. Well isn't this obvious? And isn't the need for graph development depending on the analytical goals - I am not sure if I can follow the argument here.

minor issues:

data is all available to what I could track, however, one has to dive into supplements. A clear data availability statement with instructions where to find the details is missing and needs to be added.

persephone visualization is challenging. Maybe it is comprehensive but I don't think it is necessarily intuitive. Have you tried the recent development from John Lovell at Hudson Alpha (GENESPACE - look at BiorXiv)?

I appreciate the authors use ref 46 for wheat in wheat as this is part of their own work, however, in the context still the IWGSC 2018 ref in Science is probably more appropriate? - unless you want to make a point out of impact of Hifi sequencing in wheat - which is not the case at current.

abstract: "effects on gene regulation" - no! you only report correlations between datasets, no functional proof.

intro: "almost" 10 billion by (exactly) 2064 - I find this mix of approximation and exactness curious - sure you are citing here, but ...

intro: why sequence diversity is a natural variation "tool box"??

Point-by-point response to REVIEWERS' COMMENTS

**We really appreciate the reviewers' comments and feel the manuscript is much better**
**and more comprehensive. Following their comments and the editor's recommendation,**
**we have written a major revision that addresses the reviewer's comments below:**

**Reviewer #1**

**Question 1:** The authors have tried to resolve inversions in population scale by combining 18
rice high-quality assemblies. However, to my knowledge, up to 60 rice high quality
assemblies have been available so far, it's much better to combine all available assemblies
together to investigate inversions at population level.

Moreover, the authors investigated inversion distribution in 5 additional wild species with
high-quality genomes (See line 246-250), unfortunately, they were noted "unpublished data",
those wild species will enhance the quality of this study.

**Author response:** Thanks for your suggestions. This is a very good and important comment
that enhances our work.

Indeed, including our platinum standard pan-genome resources, there are three recent
papers that released designated "high-quality" genomes, *i.e.*, Qin et al., 2021, Cell; Zhang et
al., 2022, Genome Research, and Shang et al., 2022, Cell Research. The first of the three
papers used PacBio sequencing technology, while the remaining two used Oxford Nanopore
Technology (ONT). Following the reviewer's comments, we re-called inversions by adding
several newly sequenced high-quality genome from both Qin's and Zhang's studies (n=65).
However, the genome sequences from the Shang et al., 2022 paper: "*Genome sequencing*
*data of 251 accessions in this study have been deposited in the NCBI Sequence Read Archive*
*(<https://www.ncbi.nlm.nih.gov/sra>) under BioProjects PRJNA656318 and PRJNA692836*",
have yet to be publicly released (a screenshot as below).

Prior to re-calling inversions, we assessed the genome quality of the 33 genomes from
Qin's study, and the 65 genomes from Zhang's study (See line 192 - 208). First, we called
N50s on the contigs (as the scaffold N50 was reported in some cases) to give us a more
accurate representation of a contiguous assembly and we only kept genomes with Contig
N50s > 3Mbp. Secondly, we validated genome assembly correctness by genome-wide dot-
plots and found that some assemblies had scaffolding, or possibly assembly errors. These two
parameters can indicate the quality of genome assemblies and is essential when running a

meta-analysis of INVs between genomes, as poor-quality genomes could lower our ability to
detect inversions, or falsely introduce INVs when in-fact it is a genome assembly error.
Ultimately, we added 29 (out of 33) newly sequenced PacBio genomes from Qin’s study, and
28 (out of 65) newly sequenced ONT genomes from Zhang’s study into our 18-genome data
package, for a total of 75 high quality genomes.

SRA	PRJNA692836	SRA	PRJNA656318
Create alert	Advanced	Create alert	Advanced

⚠ The following term was not found in SRA: PRJNA692836.

🔍 No items found.

⚠ The following term was not found in SRA: PRJNA656318.

🔍 No items found.

In addition, we do agree that the 5 additional wild *Oryza* high-quality genomes (*i.e.* *O.*
*nivara* [AA], *O. glaberrima* [AA], *O. barthii* [AA], *O. coarctata* [KKLL] and *O. alta*
[CCDD]), will enhance our study, however we only used these genomes to validate species
specific inversions. In the future, we plan to use these genomes to investigate abiotic, biotic
resistance, and neo-domestication. Thus, for the present paper they were removed.

References:

Qin P, Lu H, Du H, et al. Pan-genome analysis of 33 genetically diverse rice accessions
reveals hidden genomic variations[J]. Cell, 2021, 184(13): 3542-3558. e16.

Zhang F, Xue H, Dong X, et al. Long-read sequencing of 111 rice genomes reveals
significantly larger pan-genomes[J]. Genome Research, 2022, 32(5): 853-863.

Shang L, Li X, He H, et al. A super pan-genomic landscape of rice[J]. Cell Research, 2022:
1-19.

Shang L, Li X, He H, et al. A super pan-genomic landscape of rice[J]. Cell Research, 2022,
32(10): 878-896.

**Question 2:** The authors should compare the inversions in this study with inversions which
had been published, such as that data in Qin et al. Cell, 2021, wherein they also identified
tons of inversions. Because the authors did not perform the comparison, so the author can
determine how many new inversions have been identified, and therefore, the description at
line 236 is not accurate: “two of which (INV060390 and INV080710) were previously
reported”.

Moreover, the descriptions in the section of “five largest inversions” are boring, it was
just a list of the distribution of these inversion, and some distribution information have been
revealed before.

**Author response:** We thank the reviewer’s comments. We revisited Qin’s paper in 2021, and
also communicated with Qin. They did not make their data available publicly, but the authors
kindly shared all their inversion data with us. In Qin’s work, they identified 718 inversions,
of which 553 were overlapped with our inversion index. Since the two studies used different
pipelines, we discussed with them to include all of their genomes in our study and their *de*
*nov*o called inversions.

In addition, we do agree with reviewer 1 that that the “five largest inversion” section is
boring, so thus we removed that section from the paper and simply described the key
messages (See line 230).

**Question 3:** In the next four sections after “five largest inversions”, I did not see any
attractive and new contents which will significantly promote rice biological and evolutionary
studies. For example: the conclusion in the section “Characterization of Transposable
Element Content within Inversions and Breakpoints” (our results reveal an enrichment of TE
related sequences both within inversions and at their breakpoints), have been reported in
several papers. Same thing for the conclusion “a marked suppression of genetic
recombination is associated with inversions” of the section “Recombination Rate and
Genomic Inversions”.

The investigation about the effect of inversions on the genes located within inverted
regions, and their expression wasn’t appropriate, the effect of other sequence variations, such
as SNP and SV, were not considered, these variations were supposed to have much bigger
and direct impacts on gene expression than inversions, whereas, they were overlooked in this
investigation, and without discussions.

Additionally, regarding the section “Phenotypic consequences of inversions: Inversion
Cluster 92”, based on the description and my understanding, it was much better to say the
consequence of SNP.

**Author response:** Yes, we agree with the reviewer, inversions and other structural variants
having an effect on phenotypes has been previously demonstrated. The novelty in this paper
is actually organismal (rice), on a global (pan-genome created for the sub-populations of

*Oryza sativa*) and population scale (incorporating 3K data set, that is a compilation of
populations representing known subpopulations), and explored the effect of inversions (a
subtype of large structural variants) on TE and gene composition. We demonstrated that TEs
are enriched at breakpoints which supports TEs as a mechanism to produce inversions, and
the frequency that they may be at a population scale.

We agree with the reviewer that a gene's expression is impacted by multiple sequence
variation, *e.g.* SNPs, InDels. As one type of structure variations, inversions could impact
gene expression in multiple ways, including both indirectly (*e.g.*, increase changes in
promotor regions or change binding sites that are affected by 3D structure of sequences), and
directly (via gene disruption). In this case, we are looking for clues of the effect of inversions
on gene expression. We compared portions of DEGs located in genome-wide regions, within
inversion regions, and 20 Kb flanking regions by a permutation test ($n = 1000$). We carried
out comparisons from four tissues, *i.e.* panicle, mature leaves, young leaves and root. The
results showed that the portion of DEGs within inversion were significantly higher than
genome-wide regions, and the portion of DEGs in flanking regions were significantly lower
than genome-wide regions (see a figure as shown below). We do agree with the reviewer that
a gene's expression might be overlooked here, and it might be worth an independent research
study, but here it provide us with clues as to the effects of inversions on gene expression.

Lastly, we agree with the reviewer that it was better to say the consequence of SNPs in
Inversion Cluster 92, so we removed "Phenotypic consequences of inversions: Inversion
Cluster 92".

**Reviewer #2**

**Question 1:** Given that inversions are the main thrust of the paper, the paper is narrowly
written, with a focus only on rice. There is lots of previous evidence – although mostly not
discussed in this manuscript – that inversions affect phenotypes, gene expression, are
mediated TEs, etc. From my perspective, it's a missed opportunity to not put this rice work
into the context of the wider plant literature (maize, tomato, grapes, evolution, etc. etc.). The
only real attempt at generality is lines 90 to 93, and a bit in the first paragraph of the
Discussion.

**Author response:** Thanks for your comments, and we agree with that. Previous studies in
many species have shown evidence for the biological consequences of inversions. In this
study, we focused our study on inversions with a large number of high-quality genomes in
rice. To generally compare our study with other species, we added a new paragraph in the
introduction section of the manuscript (See line 89-101, and they were highlighted in yellow).

**Question 2:** The new or updated genomes from rufi and punct are diploid, I think – i.e., from
not a naturally selfing lineage, like rice. How were diploid genomes treated? Details on
phasing, haplotypes and heterozygosity are lacking, but probably important (particularly if
diploid chromosomes have SVs, so that their treatment is key to inferences)

**Author response:** Thanks for your comment. Both *O. rufipogon* and *O. punctata* are
autogamous. We believe the degree of outcrossing can be ~30%. The *O. rufipogon* and *O.*
*punctata* samples used in this study were from a single single-seed decent plant. Since the
expected fixation is about 0.7, phasing should not be such an issue for the interpretation of
our data.

**Question 3:** This is admittedly a stylistic thing, but I find it awkward to list the main results at
the end of the Discussion. I'd rather know what questions and going to be asked and why.
Moreover, on careful reading, I feel as if at least #2 and #3 were overstated.

**Author response:** Thanks for your comment. You are correct as this is a stylish thing. Our
group has written a few major papers like this in the past and feel it is important to get the
message out there at the beginning. The paper now has 7 points and we kept them brief, to
the point and tried to avoid overstatement (See line 135-162).

**Question 4:** - I struggled to follow the sampling and the nomenclature (e.g., Xi-adm MH63).
It'd be nice to have a sampling table with acronyms, taxon of origin, etc.

**Author response:** Thanks for your comment. We have updated the acronyms, taxon of
origin, and full name of accessions in **Table 1** and **Supplementary Table 2**, and the
acronyms were applied though the manuscript, tables and figures.

**Question 5:** - p. 175 – I appreciate the detail given to the various workflows in the M&M to
estimate pairwise inversions. It gives the impression of great care!

**Author response:** Thanks for your comments.

**Question 6:** - Line 181 – is this statement relative to IRGSP or to Oruf and Opunct?
Generally lines 181 to 190 were pretty tough to follow, given lack of knowledge about
sampled genomes. Again, a table would be nice or more explanation.

**Author response:** Thanks for your comment. We have slightly updated the description (See
line 216-225) and there is a table to help with explanation (Supplementary Table 3).

**Question 7:** - Line 200 – I desperately would have liked to see the information in this
paragraph summarized on a phylogeny, showing species and group-specific number of
inversions on nodes. Given that this paper repeatedly touts a strong phylogenetic basis as the
rationale for sampling, the lack of a phylogeny is a somewhat glaring omission.

**Author response:** We agree with reviewer's
comment. To address the suggestion and
improve the study, we added a section (See
line 260) on the analysis of a phylogenetic tree
(see Figure 3). We hope this could help readers
to better understand our study.

**Figure 3**

**Question 8:** - I was a bit confused by Figure 1. It's nice in the sense that it shows all of the
data, but I think there are no species-specific inversions in rice (vs. other species), right? And

what does each blue bar represent? I assume each one is an inversion. It seems to me that
there are many inversions shared among the Asian Rice group XI on the right hand side of
the graph, but not circled as group specific. Perhaps it's a question of the level of detail,
because it is very hard to show all of the inversions, but unfortunately I did not find the figure
particularly useful. (I'd love to see inversions in the context of chromosomes, centromeres,
etc.)

**Author response:** Thanks for the comments. We agree with the reviewer. We have removed
figure 1 since the figure didn't help as suggested.

We re-called all inversions based on 75 genomes and following this suggestion, we added to
the study inversion distribution, hotspots, and their overlap with centromeres (See line 241-
258 and Figure 2).

**Question 9:** - Line 208 – are subpopulations groups? (as in Figure 1?)

**Author response:** In the previous version, Line 208 are describing subpopulation specific or
shared inversions. In the current version (See line 301-307), we have classified them into four
different groups: Inversions segregating in *O. sativa* (S), Inversions segregating in both *O.*
*sativa* and *O. rufipogon* (SR), *O. rufipogon* specific (R), and *O. punctata* specific or AA-fixed
(i.e., ancestral state not clear) (P) (Figure 4).

**Question 10:** - Line 216 – if I followed correctly, the 3K-RGP is a short-read dataset. It's not
clear to me how that could be used to test/confirm inversions, particularly since later the
claim is made that short read data are not useful for inferring inversions after comparing the

results of this paper to the Fuertes paper. Moreover, details of methods, numbers used, ‘high
coverage’, etc., are lacking, such that is hard to follow the basis for conclusions.

**Author response:** We agree with reviewer’s comments. We compared the inversions from
genome assembly and short reads by using overlapping genomes. In doing so, we discovered
a very high frequency of false positives when using short read data (Fuertes et al., 2019) (See
line 373-378, Supplementary Table 14). In our case, we did not use short reads to call
inversion directly. We mapped the 3K-RGP Illumina reads to the reference genome. We took
a detailed look into the alignment in the genome browser (IGV), to validate the mapping of
short reads that span the previously confirmed inversions to show a clear breakpoint (See line
317 – 330) between samples. We initially did this manually but now also applied a machine
learning approach to identify inversion events across the whole 3K-RGP data set. We added
this analysis with details into the manuscript (Supplementary Note 4).

**Question 11:** - Line 256 - TE information about breakpoints is interesting. By “analyzing TE
content across the inversion index”, is this all 1054 inversions?

**Author response:** Thanks for your question. Yes, it was for all 1,054 inversions. In this
update, we applied our analysis to the full 75 genome data set and identified 1,769 inversions.

**Question 12:** - Figure 3C. I’m confused and need some explanation in the legend. The lines in
the middle seem to be inverted, but the arrows go in the same direction. The TE (0025) seems
to be an LTR that is split in both cases. What is INT?

**Author response:** We apologize for the confusion. We have corrected the second arrow,
which shows the direction in accession Minghui 63. The transposable element (ID: Os0025)
is a long terminal repeat (LTR) retrotransposon, which include two parts that were shown in
Fig 5C (the 3C in previous version), i.e. long-terminal repeats (Os0025_LTR) and integrase
gene (Os0025_INT), which is located in one of part in internal domain region of the
retrotransposon. We updated this explanation in the Figure 5c legend.

INT refers to integrase gene in Internal Domain, please see the structure of LTRs that cited
from Alzohairy et al., 2014 (<https://doi.org/10.1071/FP13339>)

Fig. 1. Schematic structure differences between long-terminal repeat (LTR) retrotransposons (RTs) of Copia and Gypsy families. 5' gag, group-specific antigen or capsid protein gene; ap, aspartic protease gene; int, integrase gene; rt, reverse transcriptase gene; rh, ribonuclease-H gene; 3' UTR, 3' untranslated region; PBS, primer binding site; DIS, dimerisation signal; PSI, packaging signal; PPT, polypurine tract.

**Question 13:** - How many inversions were validated with bioNano, as in Figure 2? Overall,
I'm not convinced that all 1054 inversions were independently validated (as claimed in point
#2 in the Intro). I do believe bioNano, but details are lacking (how low is the resolution? How
many could be confirmed?) I believe that short-reads are useful, but details are lacking here,
too, but (again) it's hard to claim they validate inversions on the one hand but are not useful
at all on the other (e.g., Fuertes).

**Author response:** We apologize for the confusion. We did not validate all inversions based
on bionano, but only for inversions greater than 1 Mb and corrected our claims in the
introduction. Of note, we do have Bionano data for 12 genomes in the 18-genome data
package.

We clarified how we applied the Illumina reads for cross checking in Supplementary Fig.
3. Compared to Fuertes's work, we did not call inversions based on short read mapping, but
only applied the alignment patterns on the inversion breakpoints. Fuertes's work, used short
read data to call INVs, which is the biggest difference between our two studies.

**Question 14:** - Lines 285 – I'm assuming that 10.9% and 7.3% is much higher than the
genome average, but it'd be nice to have an explicit comparison to the genome average to
drive this point home.

**Author response:** Yes, we agree with the reviewer. Following your suggestion, we
performed the DEG analysis in 4 independent tissues (panicle, root, young leaf and mature
leaf), and compared MH63 and ZS97 to the IRGSP RefSeq. We carried out a resampling test
(n=1000) by comparing the portion of DEGs in genome-wide regions (20 Kb and 100 Kb

window), within inversion regions, and flanking regions (20 Kb). From the results, we
 observed that the portion of DEGs within inversions are significantly higher than that found
 in genome-wide regions.

**Question 15:** - About the inversion rate estimates in the Discussion (line 411 and following).

It may be that I'm misreading things, but I think they are generally wrong and perhaps
 horribly so. Here's why. There may be 22 post-inversion events in rice, but (if one thinks in
 phylogenetic terms) there are many many more years accumulated across the rice lineages on
 the tree than 10,000. As a brief example, let's assume (for simplicity) that 17 rice genomes
 diverged 10,000 years ago. If that were true, then the numerator in the rate calculation should
 be 17*10,000 years, not 10,000 years, so that estimate is inflated about 17-fold. Of course,
 we don't know exactly know when each of the separate rice genomes diverged from one
 another, so the estimate of 17*10,000 years is too many. But hopefully the point is made that
 the calculations reported in this section may be way off and that the problem may require
 some consideration of population genetics given the sample.

**Author response:** We agree that this part of discussion was lacking clarity and regret that it
 was interpreted not as we intended. Indeed, if the estimate in question was based on 17 rice
 genomes, it would be wrong. What we did was based on only two genomes (Nipponbare
 (IRGSP) and KetanNangka (KN), both *Geng/Japonica (GJ)* genomes), and the number
 presented (22) is the number of inversions between these two genomes excluding any

inversion that could be found in any other from 15 genomes, *i.e.*, we are only using
inversions private to the two genomes used. This is to avoid counting inversions in regions
that could be introgressions from *Xian/Indica (XI)*, as the time to the most recent common
ancestor (MRCA) between *Xian/Indica* and *Geng/Japonica* goes much farther than 10k years.
This approach was in fact over-cautious because it is reasonable that *KetanNangka* would
share some inversions with other japonica genomes.

Our inversion rate estimates were calculated as the (number of inversions between 2
genomes) / (2*estimated time to coalescence *i.e.* 10kya). Since this *O. sativa*-only estimate
was much higher than the two preceding estimates, in the revised version we are giving a
more detailed analysis, by

- 1) using a larger collection of *GJ* genomes (14) to base our estimates on a larger set of
comparisons (out of 75 genomes),
- 2) measuring divergence in the inversion regions using SNP data in order to test whether
any of the remaining inversions could have been introgressed from *XI* populations in the past
but did not make it into the representative genomes.
- 3) Adjusting divergence time used based on recent work (Gutaker et al., 2020) and
theoretical considerations, using a more conservative estimate of 14,200 years to allow time
for coalescence within the ancestral population.

This leads to a more conservative estimate, which is closer to the one based on cross-species
comparison, however is still higher than cross-species one. We want to note however, that
estimates based on inter-species comparisons are likely serious underestimates as detecting
inversions becomes more difficult with higher divergence - which can be already seen
between diverged populations of *O. sativa*.

Additionally, under larger divergence times, there may be under-counted due to the
recurrence of inversions (inversion can occur in same region). In a recent study in human
(Porubsky et al., 2022), the authors estimated a recurrence rate of 2.7 inversions per 10k
generations per locus (at certain loci), which is slightly less than half of our genome-wide
rate.

We thank the reviewer for making this point and have updated the methods, results and
discussion accordingly (See line 269 - 298).

- 1. Porubsky D, Höps W, Ashraf H, et al. Recurrent inversion polymorphisms in humans
associate with genetic instability and genomic disorders[J]. Cell, 2022, 185(11): 1986-2005.
e26.
- 2. Gutaker R M, Groen S C, Bellis E S, et al. Genomic history and ecology of the
geographic spread of rice[J]. Nature plants, 2020, 6(5): 492-502.

**Question 16:** - Line 432 – the comparison to Fuertes. I'd have to read that paper carefully to
see how it was done, but the title implies there were 3,000 individuals. That suggest, I think,
that the entire dataset was used as evidence to support or not the inference of inversions. I'd
expect them to have found many more inversions, but that the inversions in this paper would
be a subset of their total set. I think that'd be a more fair comparison. That said, it is indeed
puzzling that there is only 194 out of 1054 that overlapped! While I agree with the authors
that short read data is certainly less accurate than long-read data, it does make one wonder
about the accuracy of the 18 assemblies and whether there is no only errors in Fuertes but
also in assemblies that mislead inversion inference.

**Author response:** Indeed, the 3K-RGP study (Fuertes et al., 2019) listed 1,255,033
inversions, but they were identified along with other structure variations, and were listed
separately for each accession, i.e., with redundancy. Due to the limitation of detecting
inversions with short-reads, the inversions were false positive errors (Type I error) and were
not validated. Taking advantage of the overlaps with our dataset, we determined that about
90% of inversion reported by Fuentes et al. (2019) could be false. In our study, we used
dotplots to validate and long reads for correcting the inversion identification. Of note, we also
found that even with whole a genome alignment strategy, that raw inversions from any caller
should be validated to obtain precise inversions (~75%).

Following your suggestion, we compared our 1,769 inversions with the all inversions
(entire dataset) that generated by 3K-RGP (Fuentes et al. 2019), and we found 293 were
overlapped with previous report. The details are reported in Supplementary Table 4.

**Question 17:** - One time consuming but convincing way to validate inversions is to find long
reads that span the inferred junctions. I don't think that was done her, but it would certainly
go a long way to confirming the inversion inferences more convincingly.

**Author response:** Thanks for your comments. Following the reviewer's suggestion, we
validated a subset of 264 random inversion with long-read data and found that 97.73% of the
inversions could be supported at both the left and right breakpoints in the reference and
queried genomes. We updated this in the manuscript (See line 197-201) and with the
following table (Supplementary Table 5).

Genome	Number of INVs	Number of INVs could be supported by PacBio LongReads
LaMu	78	75
NaBo	74	74
CMeo	35	35
MH63	77	74
Total	264	258 (97.73%)

**Reviewer #3**

Zhou et al. present a pan-genome analysis of the major sub-populations of asian rice and two
wild rice species represented by a set of 18 whole genome assemblies. the two wild rice
assemblies are new and are provided with this manuscript. The other assemblies have been
published before.

**Question 1:** The authors main point of analysis is the cataloging of inversions larger than 100
383 bp and contextualizing these with data on recombination, LD, selection and gene expression.
As the authors point out, the detection of inversions (also larger) is not novel for plant
genomes (maybe more references to recent pan-genome studies in cereals would be justified
here?), however, here maybe a first comprehensive catalog for representatives of a crop
species' subpopulations is provided.

**Author response:** Yes, mostly, the inversions were seldom studied in detail. Our study is a
first comprehensive identification, validation, and detailed study in crop species at
subpopulation level. We also added more references on past studies of INVs and their
downstream effects to comprehensively survey the literature and results of previous work, in
rice, cereals and beyond.

**Question 2:** The study reads well and is well presented. The data displays sometimes are
rather basic - often just direct exports out of commercial analysis and vizualisation software?
Intuitivity of the figure displays could be improved - maybe?

**Author response:** Thanks for the reviewer's support and suggestions. We have improved our
image designs, and we limited visualization outputted commercial software. The only
commercial software used are ones that do not have an equivalent open-source visualization
*i.e.*, bionano maps.

**Question 3:** The study adresses an obvious point in comparative genomics as more genomes
of the same species become available. The authors are well aware of artifacts introduced into
analyses when different assembly qualities affect the analysed assemblies. Therefore, I was
surprised, given the today's costs for making a Hifi assembly for rice (haploid assembly
consumable costs in the few hundred dollars range!) that the authors did not make an effort
here to have really absolutely comparable datasets. This may only incrementally change the

presented results, however, it is a weakness of the study that could have been avoided with
modest effort and investment. The same argument applies to the annotation which was done
with the same pipeline only for the Asian rice.

**Author response:** Thanks for these great comments. Yes, indeed, Hifi assemblies have
greatly enhanced the number and quality of genome for rice and many species. As a genome
size with 400 Mb, rice could be the model crop for those species as well, which can also be
cost effective. Luckily, the rice community has contributed a large number of assembled
genomes during the last two years, which helped us to obtain a comparable dataset. In this
case, we collected all publicly available high-quality genomes and performed a permutation
simulation ($n = 1000$) to determine the number of genomes we needed to interrogate to reach
near inversion saturation. In doing so, we found that 60 high-quality genomes would be
required to have a 99% chance of identify the majority inversions with allele frequencies of 2
or greater (Figure 1).

**Question 4:** The spectrum of inversion distribution: the authors say they are randomly
distributed - Ext. figure 3 should support this claim. While the figure legend is not conclusive
and the figure itself is basic, I think the authors should have made an effort define size classes
and redo the genome distribution scan. Is it true that all sizes of inversions are evenly
distributed along the chromosomes? This is counterintuitive.

**Author response:** We have re-analyzed the spectrum of inversion distributions and adjusted
the manuscript accordingly (See line 246-252). We found that inversions were evenly
distributed except on all rice chromosomes ($n=12$) with the exception of chromosome 4
(Figure 2). Following your suggestion, we split divided the inversions into 4 groups, i.e. <1

Kb, 1-5 Kb, 5-10 Kb, and >10 Kb, and found that only >10 Kb inversions on chromosome 4
 contributed to uneven distributed (Supplementary Fig. 2).

a. < 1 Kb, b. 1 - 5 Kb, c. 5 - 10 Kb, d. > 10 Kb.

**Question 5:** The authors used the 3K rice data to genotype for the presence of inversions,
 however, they performed this only on a selected set of genotypes representing the 15 sub-
 populations. Is it really computationally so intense to include all 3000? Or is this a problem
 of sequence coverage in a certain proportion of the 3K dataset? I would love to see the
 analysis on all 3000! I wonder whether the authors tried to use the inversion catalog as a
 proxy to model and detect additional inversions in the full 3K dataset as the 15 genotypes
 sequenced will for sure not give the full pan-genome inversion spectrum. The authors outline
 the technical feasibility in the discussion. Furthermore, it would be very instructive to give an
 estimate for WGS coverage to robustly scan for the presence of inversions at population
 scale?

 **Author response:** We agree with reviewer, and yes it would be nice if we could scan the
 entire 3K-RGP dataset. As discussed above, when compared the inversion results that used
 both short read and long read data, 90% of the short-read inversion call appeared to be false
 positives. However, we also found that the alignment in IGV could hint at the presence of
 inversions if we know the coordinates.

Since the reviewer suggested that it would be very instructive to give an estimate for
 coverage to robustly scan for the presence of inversions at population scale, and to see the
 analysis on all 3K-RGP samples, we applied a machine learning approach. To do this, we
 started by manually curating 872 inversions (*O. sativa* specific) across 45 accessions
 (overlapped with short reads and long reads), and used these 39,240 inversion events to train
 a machine learning model (online method). This model was used to study the 872 inversions
 across the remaining 2,979 samples, with 5-fold cross validation. In doing so, we were able to
 assess the presence or absence of all 872 inversion at the 3K-RGP population level
 (Supplementary Note 4).

 From the estimation, we found only slightly negative correlate ($df = 3020$, p -value $< 2.2e$ -
 16 , $cor = -0.1955148$) with WGS sequence coverage and inversion validation at the
 population level (as below).

**Question 6:** TE landscape was analysed in context of inversions and it was detected that
inversions are enriched with TE content in rice. The analysis of inversions could go
further here. Have you systematically assessed the TE at the Inversion junctions? This
could reveal patterns of the mechanistic involvement for the occurrence of inversions in
the rice genome.

**Author response:** We agree with reviewer. We updated the TE related analysis, and add the
results and discussion of patterns of mechanisms for the occurrence of inversions (See
line 367-378 and Supplementary Table 10).

**Question 7:** Gene expression and inversion: the authors showed examples where inversions
and expression levels of genes in inverted regions and their orthologs in "non-inverted"
haplotypes. The data is interpreted as inversions are causal, which is not unlikely,
however, the authors do not show any functional validation (which is not trivial) and
analysis and they also do not provide complementary datasets (methylation, HiC,
ATAC etc.) which would support their hypothesis and interpretations. I would also
recommend to expand the analysis to genes adjacent or in neighborhood but not
directly affected by the inversions - regulatory sequence context can and will be
affected also for such genes and one should see a gradient perhaps?

**Author response:** Thanks for these comments. We agree with the reviewer that that
functional validation could support the hypotheses. In this paper, we are aiming to show clue
that inversions could affect gene expression.

Following your suggestion, we explored the gene expression neighborhoods, *i.e.*, flanking
 regions of inversions. We studied the portion of DEGs in flanking regions (as shown below,
 within inversions, and genome-wide regions (See the two charts below showing gene
 expression patters in 20 Kb (left) and 100 Kb (right) flanking regions. To compare the
 portion of DEGs in these regions, we performed a permutation test (n = 1000) in four tissues
 (panicle, mature leaf, young leaf and root). From the charts below, we observed that the
 portion of DEGs within inversions was scientifically higher than genome-wide regions, and
 the portion of DEGs in flanking regions was significantly lower than in genome-wide
 regions. These results demonstrate that gene expression in flanking regions of inversions
 could be affected directly or indirectly by inversions themselves, as the reviewer commented.

**Question 8:** Similarly, the interpretation of positive selection in inverted or non-inverted
 haplotypes should maybe presented and discussed with more caution.

**Author response:** Since we removed the analysis of cluster inversion 92, we deleted this part
 of the manuscript.

**Question 9:** The authors make a point in the discussion that not merging sequences into a
 pan-genome graph was an advantage here. Well isn't this obvious? And isn't the need for

graph development depending on the analytical goals - I am not sure if I can follow the
argument here.

**Author response:** Sorry for the confusion here. Yes, this is obvious. Since the pangenome
idea is one of the most popular research topics nowadays, and there are a lot of tools or
approaches that have been developed for building pangenomes. However, there are no
computation tools that cover all structure variations, especially inversions. If we want retain
all genetic diversity, we recommend the avoidance of computational tools that collapse
genomes for building pangenomes.

**Question 10:** data is all available to what I could track, however, one has to dive into
supplements. A clear data availability statement with instructions where to find the details is
missing and needs to be added.

**Author response:** Thanks for this comment. We made a website to describe the motivation
of the project and the related datasets. Please see the following link which was updated in
the manuscript <https://yongzhou2019.github.io/Rice-Population-Reference-Panel/>. We hope
this will help readers to easily track all available data used in this study.

**Question 11:** persephone visualization is challenging. Maybe it is comprehensive but I don't
think it is necessarily intuitive. Have you tried the recent development from John
Lovell at Hudson Alpha (GENESPACE - look at BiorXiv)?

**Author response:** Thanks for this comment. We tried GENESPACE, and it is a great tool
indeed, especially for orthologs and synteny analysis. Normally we do static analysis of
syntenic orthogroups in GENESPACE in most analyses. However, Persephone is a multi-
genome browser, and this is dynamic but not static like GENESPACE. Of note, we also use
Persephone as a platform to share information with the community who is interested in our
data. In this case, we uploaded our 18-genome data package in Persephone including the
genomes, gene annotations, TEs and structure variations, which are all available through this
link <https://web.persephonesoft.com/>.

**Question 12:** I appreciate the authors use ref 46 for wheat in wheat as this is part of their own
work, however, in the context still the IWGSC 2018 ref in Science is probably more

appropriate? - unless you want to make a point out of impact of Hifi sequencing in wheat -
which is not the case at current.

**Author response:** We added two papers that reported by IWGSC who reported the large
genome of wheat as well, please see below:

47. Consortium, I.W.G.S. et al. Shifting the limits in wheat research and breeding using a
fully annotated reference genome. 361, eaar7191 (2018).

48. Consortium, I.W.G.S. et al. A chromosome-based draft sequence of the hexaploid
bread wheat (*Triticum aestivum*) genome. 345, 1251788 (2014).

**Question 13:** abstract: "effects on gene regulation" - no! you only report correlations between
datasets, no functional proof.

**Author response:** We agree with the comments and changed to gene expression.

**Question 14:** intro: "almost" 10 billion by (exactly) 2064 - I find this mix of approximation
and exactness curious - sure you are citing here, but ...

**Author response:** We have modified the text to "Since the world population is expected to
increase to approximately 10-billion by 2060-2070"

**Question 15:** intro: why sequence diversity is a natural variation "tool box"??

**Author response:** We deleted this sentence.

Reviewers' Comments:

Reviewer #1:

Remarks to the Author:

A super pan-genomic landscape of rice had been published at 12 July 2022 (Shang et al., Cell Research, 2022), leading that the novelty of this manuscript is greatly decreased.

I noted that the description regarding the biological significance (in terms of gene expression) of inversion in manuscript was different with that in rebuttal document, only the gene expression within inversion was described in manuscript, whereas, the effects on gene within, nearby, at genomic scale was described in rebuttal document. This mistake should not happened.

In the section of "Characterization of gene content within inversions and their breakpoints", the authors used expression datasets of only two indica accessions to try to show the effect of inversion on gene expression. In my view, two samples were few for the effect analysis on gene expression of inversion. Additionally, the authors detected only ~10 genes within inversion were differentially expressed in each tissue between two accessions, given that the effect of other variations on gene expression were not excluded, therefore, the inversion effect on gene expression needs to reconsidered.

Reviewer #3:

Remarks to the Author:

The manuscript has been greatly improved by addressing all reviewer comments which required to include more data and re-run a substantial number of analyses. I am happy with the changes and have no further requests. Congratulations to a comprehensive and informative piece of rice genomics research.

Reviewer #4:

Remarks to the Author:

The authors generated two de novo genomes for two wild rice species: *O. rufipogon* and *O. punctata*, respectively, and analyzed the data with ~70 previously published genomes to study inversion in Asian rice. I think the manuscript is clearly written. However, I don't think the manuscript reached the novelty requirements of NC. I also have the following major and minor comments for the authors.

Major:

First, the rice pan-genome has been assembled in at least five studies started from short or long reads with/without the use of outgroups. To be honest, I didn't get why the authors assembled another pangenome with a similar design.

At the same time, the inversions have been investigated using short reads, long reads, and assemblies. I didn't see many new insights compared with previous studies.

For the two genomes the authors generated for wild rice, I guess the wild samples have high heterozygosity. In this case, diploid assemblies are expected.

The presentation of the figures is unclear, sometimes, hard to follow.

Minor:

L84-88: I didn't get this. Did the authors mean SNPs are SVs?

L91: Almost all the rice pangenome papers have analyzed the inversions.

L94-101: There are lots of good examples in plants too.

L122-124: How about the genome research and cell research papers?

L140-144: I am sure this is not right. The bias in sampling leads to such biased results since rufipogon has much higher genetic diversity.

L149-151: The authors should be very careful about this estimation, more outgroups will definitely increase this estimate.

L152-157: This has been indicated in previous studies.

L158-159: This is well-known information.

L260-267: Phylogenetic analyses are expected to be done using neutral markers, I believe most of the INVs are under selection.

L269-298: The author should be very careful with such estimations, see my comments above.

L380-406: If I understand this correctly, this is novel to some extent.

Discussion: the authors should compare their results with previous publications in detail. It is hard to get what is new.

Point-by-point response

REVIEWERS' COMMENTS

**Reviewer #1** (Remarks to the Author):

**Question 1)** A super pan-genomic landscape of rice had been published at 12 July 2022
(Shang et al., Cell Research, 2022), leading that the novelty of this manuscript is greatly
decreased.

**Author response:** We address this question in the response above. In addition, the question
of novelty was overruled by the Nature editors so we will not discuss the issue of novelty
further.

**Question 2)** I noted that the description regarding the biological significance (in terms of
**gene expression**) of inversion in manuscript was different with that in rebuttal document,
only the gene expression within inversion was described in manuscript, whereas, the effects
on gene within, nearby, at genomic scale was described in rebuttal document. This mistake
should not happened.

**Author response:** We apologize for the confusion. We added the results of gene expression
within inversions, flanking regions, and genomic scale (from Line 375-382) as below:

“Based on a comparison of transcript abundance levels between Nipponbare and MH63 and
ZS97 (across four tissue types - root, panicle, young leaf, and mature leaf) we detected that 5
23 - 12 genes from MH63 and 4 - 11 genes from ZS97 within inversions, 2 - 7 genes from
24 MH63 and 2 - 4 genes from ZS97 in inverted flanking regions, and 19 - 42 genes from MH63
and 9 - 30 genes from ZS97 that were located in non-inverted randomly resampled 20Kb
regions, were differentially expressed (DEG, fold change > 2, *p* value < 0.01)
(Supplementary Fig. 7).”

**Question 3)** In the section of “Characterization of gene content within inversions and their
breakpoints”, the authors used expression datasets of only two indica accessions to try to
show the effect of inversion on gene expression. In my view, two samples were few for the
effect analysis on gene expression of inversion. Additionally, the authors detected only ~10

genes within inversion were differentially expressed in each tissue between two accessions,
given that the effect of other variations on gene expression were not excluded, therefore, the
inversion effect on gene expression needs to be reconsidered.

**Author response:** Thanks for the comments. We do agree with the reviewer's view. For
gene expression, we used data from the IRGSP RefSeq (*GJ/japonica*) and two *XI/Indica*
genomes (MH63 and ZS97), and fortunately, the RNA was deep sequenced with multiple
replicates. Regarding the differential transcript abundance, we do agree with the reviewer that
other variations, e.g., SNPs and PAVs have effects on genes' expression and addressed this
concern by stating "the possible effect of inversions" (Line 373).

**Reviewer #3** (Remarks to the Author):

The manuscript has been greatly improved by addressing all reviewer comments which
required to include more data and re-run a substantial number of analyses. I am happy with the
changes and have no further requests. Congratulations to a comprehensive and informative
piece of rice genomics research.

**Author response:** Thank you very much for your previous advice and support.

**Reviewer #4** (Remarks to the Author):

The authors generated two de novo genomes for two wild rice species: *O. rufipogon* and *O.*
*punctata*, respectively, and analyzed the data with ~70 previously published genomes to study
inversion in Asian rice. I think the manuscript is clearly written. However, I don't think the
manuscript reached the novelty requirements of NC. I also have the following major and
minor comments for the authors.

**Author response:** Thanks for the comments. The major and minor comments are really
helpful. We hope we have addressed all your questions and suggestions. As for novelty, the
question was overruled by the Nature editors so we will not discuss this concern.

Major:

**Question 1)** First, the rice pan-genome has been assembled in at least five studies started
from short or long reads with/without the use of outgroups. To be honest, I didn't get why the
authors assembled another pangenome with a similar design.

**Author response:** As discussed in the revised discussion, a total of 5 Asian rice pan-genome
studies have been published to date. Two used Illumina data and 3 PacBio, ONT or both and
only 3 studies called inversions (Table 3, as shown above). The one Illumina study showed a
91.5% false discovery rate and was thus discounted almost entirely. The Qin et al., 2021
paper sequenced 33 high quality genomes and used the same strategy we did to call
inversions. However, this paper did not interrogate the full genetic diversity of Asian rice,
encompassing only 9 subpopulations, and did not validate any of their inversions. The Shang
et al., 2022 paper generated 174 ONT genomes, however, all assemblies were reported as
contigs and not chromosome level scaffolds which can lead to serious errors in the evaluation
of inversions (please see our response to the editor above). In addition, the paper did report
the presence of 2,784 raw inversions, however no validations were reported, and, more
importantly, this data has not been made available (please see our response to the editor
above).

Regarding the last comment about design – of all the pan-genome studies, our design is the
most unique to all 5 in that we made no attempts to computationally collapse our 73-genome
data into a graph and kept all assemblies in their native state. This permitted us to precisely
identify inversions and inversion boundaries, as well as assess their frequency at the
population level using machine learning.

**Question 2)** At the same time, the inversions have been investigated using short reads, long
reads, and assemblies. I didn't see many new insights compared with previous studies.

**Author response:** Thanks for your comments.

Please see answer to question 1.

Further, for all previous pangenome reports in Asian rice, there were no attempts made to:

- 1. Validate all inversions at the individual genome and population level
- 2. Investigate the effects of inversions on recombination, LD and gene expression
- 3. Calculate inversion rates at the species level - i.e., *O. sativa* vs, *O. rufipogon*, vs. *O.*
*punctata*.

**Question 3)** For the two genomes the authors generated for wild rice, I guess the wild
samples have high heterozygosity. In this case, diploid assemblies are expected.

**Author response:** Thanks for the comments. In the case of both *O. rufipogon* and *O.*
*punctata*, while they are wild species, they are predominantly selfing. Out-crossing rates for
the two species are on the order of ~25%. Additionally, the *O. rufipogon* accession IRGC
106523 was acquired 1991-07-30 and the *O. punctata* accession IRGC 105690 on 1987-04-
28. Both were maintained over a number of generations since acquisition by the IRRI
genebank obtaining seed from bagged panicles. Seed from these cycles of regeneration would
be expected to be selfed and lead to more fixation within the accession. Further, DNA was
obtained from a single plant of each for production of the genome builds.

**Question 3)** The presentation of the figures is unclear, sometimes, hard to follow.

**Author response:** Thanks for the comments. We shared and discussed all figures with
authors and colleagues again, and made edits based on their suggestions. We hope the figures
are now easier to follow.

Minor:

**Question 4)** L84-88: I didn't get this. Did the authors mean SNPs are SVs?

**Author response:** Thanks for the point. We have modified this in the manuscript (from Line
84-88), and consider both SNPs and structure variations (SVs, i.e., INs/DELs, TRAs, and
INVs) as "standing natural variation", as below:

“One source of the raw material required to meet this urgent demand is the standing natural
variation that exists in the genomes of the more than 500,000 accessions of rice and its wild
relatives deposited in germplasm banks around the world², i.e., single nucleotide
polymorphisms [SNPs], insertions/deletions [INs/DELs], translocations [TRAs], and
inversions [INVs].”

**Question 5)** L91: Almost all the rice pangenome papers have analyzed the inversions.

**Author response:** Thanks for the comment.

Please see answer to Question 2

There are three recent papers that have reported the inversions, i.e., Fuentes et al., 2019, Qin
et al., 2021 and Shang et al., 2022 (Cell Research) (Table 3, as shown above). We compared
with the first of two papers in detail and discuss in the discussion, however, the Shang et al.
(2022) paper has not released their data (i.e., inversion data, sequences and annotations [see
our reply above], and contigs [fragmented genomes] so it was impossible to assess (please
see our response to the editor, point 6).

**Question 6)** L94-101: There are lots of good examples in plants too.

**Author response:** Thanks for the great comment.

We modified the manuscript accordingly by adding some of important studies in plants (lines
101-103), as below:

“In plants, INVs have been reported to play roles in, for example - local adaptation^{26,27},
genome-environment associations²⁷, gene regulation^{26,28,29}, flowering time²⁸, seed
germination²⁸, and fruit shape²⁹.”

**Question 7)** L122-124: How about the genome research and cell research papers?

**Author response:** Thanks for the question. The Zhang et al., 2021 (genome research paper)
didn't mention any inversion studies, and only PAV related data are available. The Shang et
al., 2022 (cell research paper) mentioned inversions, but the data was not released and the
quality of the genomes (in contigs only) are not well suited for inversion studies (as discussed
in Question 5, and our response to the editor, point 6).

**Question 8)** L140-144: I am sure this is not right. The bias in sampling leads to such biased
results since rufipogon has much higher genetic diversity.

**Author response:** Thanks for the comment. We did not use a diversity panel of *O. rufipogon*
in this case, so we did not consider the genetic diversity of *O. rufipogon* here. We reported
this and that might be true based our pan-genome index result.

**Question 9)** L149-151: The authors should be very careful about this estimation, more
outgroups will definitely increase this estimate.

**Question 10)** L269-298: The author should be very careful with such estimations, see my
comments above.

**Author response:** Thanks for these comments. Since these two comments are related, so we
combined them into one reply.

Since we had somewhat different estimates from different outgroups, it may indeed make
sense to rephrase the summary more carefully, e.g., instead of using a point estimate (700
inversions/MY) we modified it as 735 – 749 inversions/MY.

We assume that we have already used outgroups from a “wide phylogenetic range” - from
one very close to *GJ*-tmp (like *GJ*-trop) to *XI/indica* to *O. rufipogon* to *O. punctata*. One
minor thing in using other *O. rufipogon* references is the common introgression from *O.*
*sativa* to *O. rufipogon*. At the same time, it might be difficult to use more outgroups for
genome-wide comparison. Going farther beyond *O. punctata* may result in too few
alignments (due to the sequence similarity) and seriously underestimate the inversion rate.

**Question 11)** L152-157: This has been indicated in previous studies.

**Question 12)** L158-159: This is well-known information.

**Author response:** We thank the reviewer for these comments. Since these two comments are
related we combined them into a single reply.

We agree that our results confirm prior work that TEs are associated with inversions and their
breakpoints. We pointed this out in the manuscript, in results Line 336, and wrote:

“Transposable elements (TEs) are known to be associated with inversions^{11,44}”

**Question 13)** L260-267: Phylogenic analyses are expected to be done using neutral makers, I
believe most of the INVs are under selection.

**Author response:** Thanks for the comment. In this case we would like to point out that
inversions could also be applied for phylogenic analysis, since the result was almost identical
with our previous analysis using SNPs.

**Question 14)** L380-406: If I understand this correctly, this is novel to some extent.

**Author response:** Thanks very much for your comment.

**Question 15)** Discussion: the authors should compare their results with previous publications
in detail. It is hard to get what is new.

**Author response:** We apologize the confusion. We actually compared our results with
previous publications, e.g., Fuentes et al., 2019 and Qin et al., 2021, which was described in
lines 199-205, as below:

“Our inversion index was compared with previous studies in rice, e.g., the 3K-RGP¹¹ study
and the pan-genome analysis of 33 rice genomes¹². Inversions were treated as “identical” if
they matched the following two criteria: 1) the inversion length difference was smaller than
200 bp, and 2) the differences between the coordinates of breakpoints of inversions were
smaller than 100 bp. In doing so, we found that of the 1769 nonredundant inversions
identified, 38.6% were previously identified (Supplementary Data 4).”